# Finding significant combinations of features in the presence of categorical covariates

**Laetitia Papaxanthos**\*, **Felipe Llinares-López**\*, **Dean Bodenham, Karsten Borgwardt**
Machine Learning and Computational Biology Lab
D-BSSE, ETH Zurich

*Equally contributing authors.

## Abstract

In high-dimensional settings, where the number of features $p$ is much larger than the number of samples $n$, methods that systematically examine *arbitrary combinations* of features have only recently begun to be explored. However, none of the current methods is able to assess the association between feature combinations and a target variable while conditioning on a categorical covariate. As a result, many false discoveries might occur due to unaccounted confounding effects.

We propose the Fast Automatic Conditional Search (FACS) algorithm, a significant discriminative itemset mining method which conditions on categorical covariates and only scales as $O(k \log k)$, where $k$ is the number of states of the categorical covariate. Based on the Cochran-Mantel-Haenszel Test, FACS demonstrates superior speed and statistical power on simulated and real-world datasets compared to the state of the art, opening the door to numerous applications in biomedicine.

## 1 Introduction

In the last 10 years, the amount of data available is growing at an unprecedented rate. However, in many application domains, such as computational biology and healthcare, the amount of features is growing much faster than typical sample sizes. Therefore, statistical inference in high-dimensional spaces has become a tool of the utmost importance for practitioners in those fields. Despite the great success of approaches based on sparsity-inducing regularizers [16, 2], the development of methods to systematically explore arbitrary combinations of features and assess their statistical association with a target of interest has been less studied. Exploring all combinations of $p$ features is equivalent to handling a $2^p$-dimensional space, thus combinatorial feature discovery exacerbates the challenges for statistical inference in high-dimensional spaces even for moderate $p$.

Under the assumption that features and targets are binary random variables, recent work in the field of *significant discriminative itemset mining* offers tools to solve the computational and statistical challenges incurred by combinatorial feature discovery. However, all state-of-the-art approaches [15, 10, 13, 7, 8] share a key limitation: no method exists to assess the *conditional* association between feature combinations and the target. The ability to condition the associations on an observed covariate is fundamental to correct for *confounding* effects. If unaccounted for, one may find many false positives that are actually associated with the covariate and not the class of interest [17]. For example, in medical case/control association studies, it is common to search for combinations of genetic variants that are associated with a disease of interest. In this setting, the class labels are the health status of individuals, sick or healthy. The features represent binary genetic variants, encoded as 1 if the variant is altered and as 0 if not. Often, in high-order association studies, a subset of genetic variants are combined to form a binary variable whose value is 1 if the subset only contains altered genetic variants and is 0 otherwise. A subset of genetic variants is associated with the class label if the frequencies of altered combinations in each class are statistically different. However, it is often the

case that the studied samples belong to several subpopulations, for example African-American, East Asian or European Caucasian, which show differences in the prevalence of some altered combinations of genetic variants because of systematic ancestry differences. When, additionally, the subpopulations clusters are unevenly distributed across classes, it can result in false associations to the disease of interest [12]. This is the reason why it is necessary to model ancestry differences between cases and controls in the presence of population structure or to correct for covariates in more general settings.

*Hence our goal in this article is to present the first approach to significant discriminative itemset mining that allows one to correct for a confounding categorical covariate.*

To reach this goal, we present the novel algorithm `FACS`, which enables significant discriminative itemset mining with categorical covariates through the Cochran-Mantel-Haenszel test [9] in $O(k \log k)$ time, where $k$ is the number of states of the categorical covariate, compared to the standard implementation which is exponential in $k$.

The rest of this article is organized as follows: In Section 2 we define the problem to be solved and introduce the main theoretical concepts from related work that `FACS` is based on, namely the Cochran-Mantel-Haenszel-test and Tarone's testability criterion. In Section 3, we describe in detail our contribution, the `FACS` algorithm and its efficient implementation. Finally, Section 4 validates the performance of our method on a set of simulated and biomedical datasets.

## 2 Problem statement and related work

In this section we introduce the necessary background, notation and terminology for the remainder of this article. First, in Section 2.1 we rigorously define the problem we solve in this paper. Next, in Sections 2.2 and 2.3 we describe two key elements on which our method is based: the Cochran-Mantel-Haenszel (CMH) test and Tarone's testability criterion.

### 2.1 Discovering significant feature combinations in the presence of a categorical covariate

We consider a dataset of $n$ observations $\mathcal{D} = \{(\mathbf{u}_i, y_i, c_i)\}_{i=1}^{n}$, where the $i$th observation consists of: (I) a feature vector $\mathbf{u}_i$ consisting of $p$ binary features, $u_{i,j} \in \{0, 1\}$ for $j = 1, \ldots, p$; (II) a binary class label, $y_i \in \{0, 1\}$; and (III) a categorical covariate $c_i$, which has $k$ categories, i.e. $c_i \in \{1, 2, \ldots, k\}$. Given any subset of features $\mathcal{S} \subseteq \{1, 2, \ldots, p\}$, we define its induced *feature combination* for the $i$th observation as $z_{i,\mathcal{S}} = \prod_{j \in \mathcal{S}} u_{i,j}$, such that $z_{i,\mathcal{S}}$ takes value 1 if and only if $u_{i,j} = 1$ for all features in $\mathcal{S}$. Now, we use $Z_{\mathcal{S}}$ to denote the feature combination induced by $\mathcal{S}$, of which $z_{i,\mathcal{S}}$ is the realization for the $i$th observation. Similarly, we use $Y$ to denote the label, and $C$ to denote the covariate, of which $y_i$ and $c_i$ are realizations, respectively, for $i = 1, 2, \ldots, n$. Below we use the standard notation $A \perp\!\!\!\perp B$ to denote "$A$ is statistically independent of $B$".

Typically, significant discriminative itemset mining aims to find *all* feature subsets $\mathcal{S}$ for which a statistical association test rejects the null hypothesis, namely $Z_{\mathcal{S}} \perp\!\!\!\perp Y$, after a rigorous correction for multiple hypothesis testing. However, for any feature subset such that $Z_{\mathcal{S}} \not\perp\!\!\!\perp Y$ but $Z_{\mathcal{S}} \perp\!\!\!\perp Y \mid C$, the association between $Z_{\mathcal{S}}$ and $Y$ is exclusively mediated by the covariate $C$, which acts in this case as a *confounder* creating spurious associations.

**Our goal:** *In this work, the aim is to find all feature subsets $\mathcal{S}$ for which a statistical association test rejects the null hypothesis $Z_{\mathcal{S}} \perp\!\!\!\perp Y \mid C$, thus allowing to correct for a confounding categorical covariate while keeping the computational efficiency, statistical power and the ability to correct for multiple hypothesis testing of existing methods.*

In the remainder of this section we will introduce two fundamental concepts our work relies upon. The first one is the Cochran-Mantel-Haenszel (CMH) test, which offers a principled way to test if a feature combination $Z_{\mathcal{S}}$ is conditionally dependent on the class labels $Y$ given the covariate $C$, that is, to test the null hypothesis $Z_{\mathcal{S}} \perp\!\!\!\perp Y \mid C$. The second concept is Tarone's testability criterion, which allows a correction for multiple hypothesis testing while retaining large statistical power, in scenarios such as ours where billions or trillions of association tests must be performed.

Tarone's testability criterion has only been successfully applied to unconditional association tests, such as Fisher's exact test [6] or Pearson's $\chi^2$ test [11]. Thus, the state-of-the-art in significant discriminative itemset mining forces one to choose between: (a) using Bonferroni's correction, resulting in very low statistical power or an arbitrary limit in the cardinality of feature subsets (e.g. [18]), or (b) using Tarone's testability criterion, losing the ability to account for covariates and resulting in potentially many confounded patterns being deemed significant [15, 13, 7, 8].

**Our contribution:** *In this paper, we propose* `FACS`*, a novel algorithm that allows applying Tarone's testability criterion to the CMH test, allowing to correct for a categorical covariate in significant discriminative itemset mining for the first time.* `FACS` *will be introduced in detail in Section 3.*

## 2.2 Conditional association testing with the Cochran-Mantel-Haenszel (CMH) test

To test if $Z_{\mathcal{S}} \perp\!\!\!\perp Y \,|\, C$, the CMH test [9] arranges the $n$ realisations of $\{(z_{i,\mathcal{S}}, y_i, c_i)\}_{i=1}^n$ into $k$ distinct $2 \times 2$ contingency tables, one table for each possible value of the covariate $c$, as:

| Variables | $z_{\mathcal{S}} = 1$ | $z_{\mathcal{S}} = 0$ | Row totals |
|---|---|---|---|
| $y = 1$ | $a_{\mathcal{S},j}$ | $n_{1,j} - a_{\mathcal{S},j}$ | $n_{1,j}$ |
| $y = 0$ | $x_{\mathcal{S},j} - a_{\mathcal{S},j}$ | $n_{2,j} - x_{\mathcal{S},j} + a_{\mathcal{S},j}$ | $n_{2,j}$ |
| Col totals | $x_{\mathcal{S},j}$ | $n_j - x_{\mathcal{S},j}$ | $n_j$ |

where: (I) $n_j$ is the number of observations with $c = j$, $n_{1,j}$ of which have class label $y = 1$ and $n_{2,j}$ of which have class label $y = 0$; (II) $x_{\mathcal{S},j}$ is the number of observations with $c = j$ and $z_{i,\mathcal{S}} = 1$; (III) $a_{\mathcal{S}}$ is the number of observations with $c = j$, class label $y = 1$ and $z_{i,\mathcal{S}} = 1$. Based on $\{n_j, n_{1,j}, x_{\mathcal{S},j}, a_{\mathcal{S},j}\}_{j=1}^k$, a $p$-value $p_{\mathcal{S}}$ for feature combination $Z_{\mathcal{S}}$ is computed as:

$$p_{\mathcal{S}} = 1 - F_{\chi_1^2} \left( \frac{\left( \sum_{j=1}^k a_{\mathcal{S},j} - \frac{x_{\mathcal{S},j} n_{1,j}}{n_j} \right)^2}{\sum_{j=1}^k \frac{n_{1,j}}{n_j} \left(1 - \frac{n_{1,j}}{n_j}\right) x_{\mathcal{S},j} \left(1 - \frac{x_{\mathcal{S},j}}{n_j}\right)} \right) \tag{1}$$

where $F_{\chi_1^2}(\cdot)$ is the distribution function of a $\chi^2$ random variable with 1 degree of freedom. Finally, the feature combination $Z_{\mathcal{S}}$ and its corresponding feature subset $\mathcal{S}$ will be deemed significantly associated if the $p$-value $p_{\mathcal{S}}$ falls below a *corrected significance threshold* $\delta$, that is, if $p_{\mathcal{S}} \leq \delta$.

The CMH test can be understood as a form of *meta-analysis* applied to $k$ disjoint datasets $\{\mathcal{D}_j\}_{j=1}^k$, where $\mathcal{D}_j = \{(\mathbf{u}_i, y_i) \,|\, c_i = j\}$ contains only observations for which the covariate $c$ takes value $j$. For *confounded feature combinations*, the association might be large in the entire dataset $\mathcal{D}$, but small for conditional datasets $\mathcal{D}_j$. Thus, the CMH test will *not* deem such feature combinations significant.

## 2.3 The multiple testing problem in discriminative itemset mining

In our setup, one must perform $2^p - 1$ association tests, one for each possible subset of features. Even for moderate $p$, this leads to an enormous number of tests, resulting in a large *multiple hypothesis testing problem*. To produce statistically reliable results, the significance threshold $\delta$ will be chosen to guarantee that the Family-Wise Error Rate (FWER), defined as the probability of producing any false positives, is upper-bounded by a significance level $\alpha$. FWER control is most commonly achieved with Bonferroni's correction [3, 5], which in our setup would imply using $\delta = \alpha/(2^p - 1)$ as significance threshold. However, Bonferroni's correction tends to be overly conservative, resulting in very low statistical power when the number of tests performed is large. In contrast, recent work in significant discriminative itemset mining [15, 10, 13, 7] showed that, in this setting, Bonferroni's correction can be outperformed in terms of statistical power by Tarone's testability criterion [14].

Tarone's testability criterion is based on the observation that, for some discrete test statistics based on contingency tables, a minimum attainable $p$-value can be computed as a function of the table margins. Let $\Psi(\mathcal{S})$ denote the minimum attainable $p$-value corresponding to the contingency table of feature combination $Z_{\mathcal{S}}$. By definition, $p_{\mathcal{S}} \geq \Psi(\mathcal{S})$, therefore $\Psi(\mathcal{S}) > \delta$ implies that feature combination $Z_{\mathcal{S}}$ can never be deemed significantly associated, and hence it cannot cause a false positive. In other words, feature subsets $\mathcal{S}$ for which $\Psi(\mathcal{S}) > \delta$ are irrelevant as far as the FWER is concerned. In Tarone's terminology, $\mathcal{S}$ is said to be *untestable*. Thus, defining the *set of testable feature subsets at level* $\delta$ as $\mathcal{I}_T(\delta) = \{\mathcal{S} \,|\, \Psi(\mathcal{S}) \leq \delta\}$, Tarone's testability criterion obtains the corrected significance threshold as $\delta_{tar} = \max\{\delta : \text{FWER}_{tar}(\delta) \leq \alpha\}$, where $\text{FWER}_{tar}(\delta) = \delta |\mathcal{I}_T(\delta)|$. Note that this amounts to applying a Bonferroni correction to feature subsets $\mathcal{S}$ in $\mathcal{I}_T(\delta)$ only. FWER control follows from the fact that untestable feature subsets cannot affect the FWER. Since in practice $|\mathcal{I}_T(\delta)| \ll 2^p - 1$, Tarone's testability criterion often outperforms Bonferroni's correction in terms of statistical power by a large margin.

The main practical limitation of Tarone's testability criterion is its computational complexity. Naively computing $\delta_{tar}$ would involve explicitly enumerating all $2^p - 1$ feature subsets and evaluating their respective minimum attainable $p$-values, something unfeasible even for moderate $p$. Existing work in significant discriminative pattern mining solves that limitation by exploiting specific properties of

certain test statistics, such as Fisher's Exact Test or Pearson's $\chi^2$ test, that allow to apply branch-and-bound algorithms to evaluate $\delta_{tar}$. However, the properties those algorithms rely on do not apply to conditional statistical association tests, such as the CMH test. In the next section, we present in detail our novel approach to apply Tarone's method to the CMH test.

## 3 Our contribution: The FACS algorithm

This section introduces the Fast Automatic Conditional Search (`FACS`) algorithm, the first approach that allows the application of Tarone's testability criterion to the CMH test in a computationally efficient manner. Section 3.1 discusses the main challenges facing `FACS` and summarizes how `FACS` improves the state of the art. Section 3.2 provides a high-level description of the algorithm. Finally, Sections 3.3 and 3.4 detail the two key steps of `FACS`, which are also the main algorithmic contributions of this work.

### 3.1 Overview and Contributions

The main objective of the `FACS` algorithm, described in Section 3.2 below, can be summarised as:

**Objective:** *Given a dataset $\mathcal{D} = \{(\mathbf{u}_i, y_i, c_i)\}_{i=1}^n$, the goal of `FACS` is to:*

1. *Compute Tarone's corrected significance threshold $\delta_{tar}$.*
2. *Retrieve all feature subsets $\mathcal{S}$ whose p-value $p_{\mathcal{S}}$ is below $\delta_{tar}$.*

*For both (1) and (2), the test statistic of choice will be the CMH test, thus allowing to correct for a confounding categorical covariate as described in Section 2.2.*

The key contribution of our work is to bridge the gap between Tarone's testability criterion and the CMH test. Firstly, in Section 3.3, we show for the first time that Tarone's method can be applied to the CMH test. More importantly, in Section 3.4 we introduce a novel branch-and-bound algorithm to efficiently compute $\delta_{tar}$ without requiring the function $\Psi$ computing Tarone's minimum attainable $p$-value to be monotonic. This allows us not only to apply Tarone's testability criterion to the CMH test, but to do so as efficiently as existing methods not able to handle confounding covariates do.

### 3.2 High-level description of `FACS`

As shown in the pseudocode in Algorithm 1, conceptually, `FACS` performs two main operations:

---

**Algorithm 1** FACS

**Input:** Dataset $\mathcal{D} = \{(\mathbf{u}_i, y_i, c_i)\}_{i=1}^n$, target FWER $\alpha$
**Output:** $\{\mathcal{S} \,|\, p_{\mathcal{S}} \leq \delta_{tar}\}$
1: Initialize global variables $\delta_{tar} = 1$ and $\mathcal{I}_T(\delta_{tar}) = \emptyset$
2: $\delta_{tar}, \mathcal{I}_T(\delta_{tar}) \leftarrow$ `tarone_cmh`$(\emptyset)$
3: Return $\{\mathcal{S} \in \mathcal{I}_T(\delta_{tar}) \,|\, p_{\mathcal{S}} \leq \delta_{tar}\}$

---

**Algorithm 2** `tarone_cmh`

**Input:** Current feature subset being processed $\mathcal{S}$
1: **if** `is_testable_cmh`$(\mathcal{S}, \delta_{tar})$ **then** {see Section 3.3}
2:     Append $\mathcal{S}$ to $\mathcal{I}_T(\delta_{tar})$
3:     $\text{FWER}_{\text{tar}}(\delta_{tar}) \leftarrow \delta_{tar}|\mathcal{I}_T(\delta_{tar})|$
4:     **while** $\text{FWER}_{\text{tar}}(\delta_{tar}) > \alpha$ **do**
5:         Decrease $\delta_{tar}$
6:         $\mathcal{I}_T(\delta_{tar}) \leftarrow$ $\{\mathcal{S} \in \mathcal{I}_T(\delta_{tar}) : $ `is_testable`$(\mathcal{S}, \delta_{tar})\}$
7:         $\text{FWER}_{\text{tar}}(\delta_{tar}) \leftarrow \delta_{tar}|\mathcal{I}_T(\delta_{tar})|$
8: **if not** `is_prunable_cmh`$(\mathcal{S}, \delta_{tar})$ **then** {see 3.4}
9:     **for** $\mathcal{S}' \in \text{Children}(\mathcal{S})$ **do**
10:       `tarone_cmh`$(\mathcal{S}')$

---

Firstly, Line 2 invokes the routine `tarone_cmh`, described in Algorithm 2. This routine uses our novel branch-and-bound approach to efficiently compute Tarone's corrected significance threshold $\delta_{tar}$ and the set of testable feature subsets $\mathcal{I}_T(\delta_{tar})$.

Secondly, using the significance threshold $\delta_{tar}$ obtained in the previous step, Line 3 evaluates the conditional association of the feature combination $Z_{\mathcal{S}}$ of each testable feature subset $\mathcal{S} \in \mathcal{I}_T(\delta_{tar})$ with the class labels, given the categorical covariate, using the CMH test as shown in Section 2.2. Note that, according to Tarone's testability criterion, untestable feature subsets $\mathcal{S} \notin \mathcal{I}_T(\delta_{tar})$ cannot be significant and therefore do not need to be considered in this step. Since in practice $|\mathcal{I}_T(\delta_{tar})| \ll 2^p - 1$, the procedure `tarone_cmh` is the most critical part of `FACS`.

The routine `tarone_cmh` uses the enumeration scheme first proposed in [10, 13]. All $2^p$ feature subsets are arranged in an enumeration tree such that $\mathcal{S}' \in \text{Children}(\mathcal{S}) \Rightarrow \mathcal{S} \subset \mathcal{S}'$. In other words,

the children of a feature subset $\mathcal{S}$ in the enumeration tree are obtained by adding an additional feature to $\mathcal{S}$. Before invoking `tarone_cmh`, in Line 1 of Algorithm 1 the significance threshold $\delta_{tar}$ is initialized to 1, the largest value it can take, and the set of testable feature combinations $\mathcal{I}_T(\delta_{tar})$ is initialized to the empty set. The enumeration procedure is started by calling `tarone_cmh` with the empty feature subset $\mathcal{S} = \emptyset$, which acts as the root of the enumeration tree[1]. All $2^p - 1$ non-empty feature subsets will then be explored recursively by traversing the enumeration tree depth-first.

Every time a feature subset $\mathcal{S}$ in the tree is visited, Line 1 of Algorithm 2 checks if it is testable, as detailed in Section 3.3. If it is, $\mathcal{S}$ is appended to the set of testable feature subsets $\mathcal{I}_T(\delta_{tar})$ in Line 2. The FWER condition for Tarone's testability criterion is checked in Lines 3 and 4. If it is found to be violated, the significance threshold $\delta_{tar}$ is decreased in Line 5 until the condition is satisfied again, removing from $\mathcal{I}_T(\delta_{tar})$ any feature subsets made untestable by decreasing $\delta_{tar}$ in Line 6 and re-evaluating the FWER condition accordingly in Line 7. Before continuing the traversal of the tree by exploring the children of the current feature subset $\mathcal{S}$, Line 8 checks if our novel pruning criterion applies, as described in Section 3.4. Only if it does not apply are all children of $\mathcal{S}$ visited recursively in Lines 9 and 10. The testability and pruning conditions in Lines 1 and 8 become more stringent as $\delta_{tar}$ decreases. Because of this, as $\delta_{tar}$ decreases along the enumeration procedure (see Line 5), increasingly larger parts of the search space are pruned. Thus, the algorithm terminates when, for the current value of $\delta_{tar}$ and $\mathcal{I}_T(\delta_{tar})$, all feature subsets that cannot be pruned have been visited.

The two most challenging steps in `FACS` are the design of an appropriate testability criterion, `is_testable_cmh`$(\mathcal{S}, \delta)$, and an efficient pruning criterion, `is_prunable_cmh`$(\mathcal{S}, \delta)$, that circumvent the limitations of the current state of the art. These are now each described in detail.

### 3.3 A testability criterion for the CMH test

As mentioned in Section 2.3, Tarone's testability criterion has only been applied to test statistics such as Fisher's exact test, Pearson's $\chi^2$ test and the Mann-Whitney $U$ Test, none of which allows for incorporating covariates. However, the following proposition shows that the CMH test also has a minimum attainable $p$-value $\Psi_{cmh}(S)$:

**Proposition 1** *The CMH test has a minimum attainable $p$-value $\Psi_{cmh}(\mathcal{S})$, which can be computed in $O(k)$ time as a function of the margins $\{n_j, n_{1,j}, x_{\mathcal{S},j}\}_{j=1}^{k}$ of the $k$ $2 \times 2$ contingency tables.*

The proof of Proposition 1, provided in the Supp. Material, involves showing that $\Psi_{cmh}(\mathcal{S})$ can be computed from the $k$ $2 \times 2$ contingency tables corresponding to $Z_{\mathcal{S}}$ (see Section 2.2) by optimising the $p$-value $p_{\mathcal{S}}$ with respect to $\{a_{\mathcal{S},j}\}_{j=1}^{k}$ while keeping the table margins $\{n_j, n_{1,j}, x_{\mathcal{S},j}\}_{j=1}^{k}$ fixed.

### 3.4 A pruning criterion for the CMH test

State-of-the-art methods [15, 8], all of which are limited to unconditional association testing, exploit the fact that the minimum attainable $p$-value function $\Psi(\mathcal{S})$, using either Fisher's exact test or Pearson's $\chi^2$ test on a single contingency table, obeys a simple monotonicity property: $\mathcal{S} \subseteq \mathcal{S}' \Rightarrow \Psi(\mathcal{S}) \leq \Psi(\mathcal{S}')$ provided that $x_{\mathcal{S}} \leq \min(n_1, n_2)$. This leads to a remarkably simple pruning criterion: if a feature subset $\mathcal{S}$ is non-testable, i.e. $\Psi(\mathcal{S}) > \delta$, and its support $x_{\mathcal{S}}$ is smaller or equal to $\min(n_1, n_2)$, then all children $\mathcal{S}'$ of $\mathcal{S}$, which satisfy $\mathcal{S} \subset \mathcal{S}'$ by construction of the enumeration tree, will also be non-testable and can be pruned from the search space. However, such a monotonicity property does *not* hold for the CMH minimum attainable $p$-value function $\Psi_{cmh}(\mathcal{S})$, severely complicating the development of an effective pruning criterion.

In Section 3.4.1 we show how to circumvent this limitation by introducing a novel pruning criterion based on defining a monotonic *lower envelope* $\widetilde{\Psi}_{cmh}(\mathcal{S}) \leq \Psi_{cmh}(\mathcal{S})$ of the original minimum attainable $p$-value function $\Psi_{cmh}(\mathcal{S})$ and prove that it leads to a valid pruning strategy. Finally, in Section 3.4.2, we provide an efficient algorithm to evaluate $\widetilde{\Psi}_{cmh}(\mathcal{S})$ in $O(k \log k)$ time, instead of a naive implementation whose computational complexity would scale exponentially with $k$, the number of categories for the covariate. Due to space constraints, all proofs are in the Supp. Material.

#### 3.4.1 Definition and correctness of the pruning criterion

As mentioned above, existing unconditional significant discriminative pattern mining methods only consider feature subsets $\mathcal{S}$ with support $x_{\mathcal{S}} \leq \min(n_1, n_2)$ to be *potentially prun-*

*able.* Analogously, we consider as potentially prunable the set of feature subsets $\mathcal{I}_{PP} = \{\mathcal{S} \mid x_{\mathcal{S},j} \leq \min(n_{1,j}, n_{2,j}) \forall j = 1, \ldots, k\}$. Note that for $k = 1$, our definition reduces to that of existing work. In itemset mining, a very large proportion of all feature subsets will have small supports. Therefore, restricting the application of the pruning criterion to potentially prunable patterns does not cause a loss of performance in practice. We can now state the definition of the lower envelope for the CMH minimum attainable $p$-value:

**Definition 1** *Let $\mathcal{S} \in \mathcal{I}_{PP}$ be a potentially prunable feature subset. The lower envelope $\widetilde{\Psi}_{cmh}(\mathcal{S})$ is defined as $\widetilde{\Psi}_{cmh}(\mathcal{S}) = \min\{\Psi_{cmh}(\mathcal{S}') \mid \mathcal{S}' \supseteq \mathcal{S}\}$.*

Note that, by construction, $\widetilde{\Psi}_{cmh}(\mathcal{S})$ satisfies $\widetilde{\Psi}_{cmh}(\mathcal{S}) \leq \Psi_{cmh}(\mathcal{S})$ for all feature subsets $\mathcal{S}$ in the set of potentially prunable patterns. Next, we show that unlike for the minimum attainable $p$-value function $\Psi_{cmh}(\mathcal{S})$, the monotonicity property holds for the lower envelope $\widetilde{\Psi}_{cmh}(\mathcal{S})$:

**Lemma 1** *Let $\mathcal{S}, \mathcal{S}' \in \mathcal{I}_{PP}$ be two potentially prunable feature subsets such that $\mathcal{S} \subseteq \mathcal{S}'$. Then, $\widetilde{\Psi}_{cmh}(\mathcal{S}) \leq \widetilde{\Psi}_{cmh}(\mathcal{S}')$ holds.*

Next, we state the main result of this section, which establishes our search space pruning criterion:

**Theorem 1** *Let $\mathcal{S} \in \mathcal{I}_{PP}$ be a potentially prunable feature subset such that $\widetilde{\Psi}_{cmh}(\mathcal{S}) > \delta$. Then, $\Psi_{cmh}(\mathcal{S}') > \delta$ for all $\mathcal{S}' \supseteq \mathcal{S}$, i.e. all feature subsets containing $\mathcal{S}$ are non-testable at level $\delta$ and can be pruned from the search space.*

To summarize, the pruning criterion `is_prunable_cmh` in Line 8 of Algorithm 2 evaluates to `true` if and only if $\mathcal{S} \in \mathcal{I}_{PP} \Leftrightarrow x_{\mathcal{S},j} \leq \min(n_{1,j}, n_{2,j}) \forall j = 1, \ldots, k$ and $\widetilde{\Psi}_{cmh}(\mathcal{S}) > \delta_{tar}$.

### 3.4.2 Evaluating the pruning criterion in $O(k \log k)$ time

In `FACS`, the pruning criterion stated above will be applied to all enumerated feature subsets. Hence, it is mandatory to have an efficient algorithm to compute the lower envelope for the CMH minimum attainable $p$-value $\widetilde{\Psi}_{cmh}(\mathcal{S})$ for any potentially prunable feature subset $\mathcal{S} \in \mathcal{I}_{PP}$.

As shown in the proof of Proposition 1 in the Supp. Material, $\Psi_{cmh}(\mathcal{S})$ depends on the pattern $\mathcal{S}$ through its $k$-dimensional vector of supports $\mathbf{x}_{\mathcal{S}} = (x_{\mathcal{S},1}, \ldots, x_{\mathcal{S},k})$. Also, the condition $\mathcal{S}' \supseteq \mathcal{S}$ implies that $x_{\mathcal{S}',j} \leq x_{\mathcal{S},j} \forall j = 1, \ldots, k$. As a consequence, one can rewrite Definition 1 as $\widetilde{\Psi}_{cmh}(\mathcal{S}) = \min_{\mathbf{x}_{\mathcal{S}'} \leq \mathbf{x}_{\mathcal{S}}} \Psi_{cmh}(\mathbf{x}_{\mathcal{S}'})$, where the vector inequality $\mathbf{x}_{\mathcal{S}'} \leq \mathbf{x}_{\mathcal{S}}$ holds component-wise. Thus, naively computing $\widetilde{\Psi}(\mathcal{S})$ would require optimizing $\Psi_{cmh}$ over a set of size $\prod_{j=1}^{k} x_{\mathcal{S},j} = O(m^k)$, where $m$ is the geometric mean of $\{x_{\mathcal{S},j}\}_{j=1}^{k}$. This scaling is clearly impractical, as even for moderate $k$ it would result in an overhead large enough to outweigh the benefits of pruning.

Because of this, in the remainder of this section we propose the last key part of `FACS`: an efficient algorithm which evaluates $\widetilde{\Psi}(\mathcal{S})$ in only $O(k \log(k))$ time. We will arrive at our final result in two steps, contained in Lemma 2 and Theorem 2.

**Lemma 2** *Let $\mathcal{S} \in \mathcal{I}_{PP}$ be a potentially prunable feature subset. The optimum $\mathbf{x}_{\mathcal{S}'}^{*}$ of the discrete optimization problem $\min_{\mathbf{x}_{\mathcal{S}'} \leq \mathbf{x}_{\mathcal{S}}} \Psi_{cmh}(\mathbf{x}_{\mathcal{S}'})$ satisfies $x_{\mathcal{S}',j}^{*} = 0$ or $x_{\mathcal{S}',j}^{*} = x_{\mathcal{S},j}$ for each $j = 1, \ldots, k$.*

In short, Lemma 2 shows that the optimum $\mathbf{x}_{\mathcal{S}'}^{*} = \{\Psi_{cmh}(\mathbf{x}_{\mathcal{S}'}) \mid \mathbf{x}_{\mathcal{S}'} \leq \mathbf{x}_{\mathcal{S}}\}$ of the discrete optimization problem defining $\widetilde{\Psi}(\mathcal{S})$ is always a vertex of the discrete hypercube $[\![\mathbf{0}, \mathbf{x}_{\mathcal{S}}]\!]$. Thus, the computational complexity of evaluating $\widetilde{\Psi}_{cmh}(\mathcal{S})$ can be reduced from $O(m^k)$ to $O(2^k)$, where $m \gg 2$ for most patterns. Finally, building upon the result of Lemma 2, Theorem 2 below shows that one can in fact find the optimal vertex out of all $O(2^k)$ vertices in $O(k \log k)$ time.

**Theorem 2** *Let $\mathcal{S} \in \mathcal{I}_{PP}$ be a potentially testable feature subset and define $\beta_{\mathcal{S},j}^{l} = \frac{n_{2,j}}{n_j}\left(1 - \frac{x_{\mathcal{S},j}}{n_j}\right)$ and $\beta_{\mathcal{S},j}^{r} = \frac{n_{1,j}}{n_j}\left(1 - \frac{x_{\mathcal{S},j}}{n_j}\right)$ for $j = 1, \ldots, k$. Let $\pi_l$ and $\pi_r$ be permutations $\pi_l, \pi_r : [\![1,k]\!] \mapsto [\![1,k]\!]$ such that $\beta_{\mathcal{S},\pi_l(1)}^{l} \leq \cdots \leq \beta_{\mathcal{S},\pi_l(k)}^{l}$ and $\beta_{\mathcal{S},\pi_r(1)}^{r} \leq \cdots \leq \beta_{\mathcal{S},\pi_r(k)}^{r}$, respectively.*

*Then, there exists an integer $\kappa \in [\![1,k]\!]$ such that the optimum $\mathbf{x}_{\mathcal{S}'}^{*} = \arg\min_{\mathbf{x}_{\mathcal{S}'} \leq \mathbf{x}_{\mathcal{S}}} \Psi_{cmh}(\mathbf{x}_{\mathcal{S}'})$ satisfies one of the two possible conditions: (I) $x_{\mathcal{S}',\pi_l(j)}^{*} = x_{\mathcal{S},\pi_l(j)}$ for all $j \leq \kappa$ and $x_{\mathcal{S}',\pi_l(j)}^{*} = 0$ for all $j > \kappa$ or (II) $x_{\mathcal{S}',\pi_r(j)}^{*} = x_{\mathcal{S},\pi_r(j)}$ for all $j \leq \kappa$ and $x_{\mathcal{S}',\pi_r(j)}^{*} = 0$ for all $j > \kappa$.*

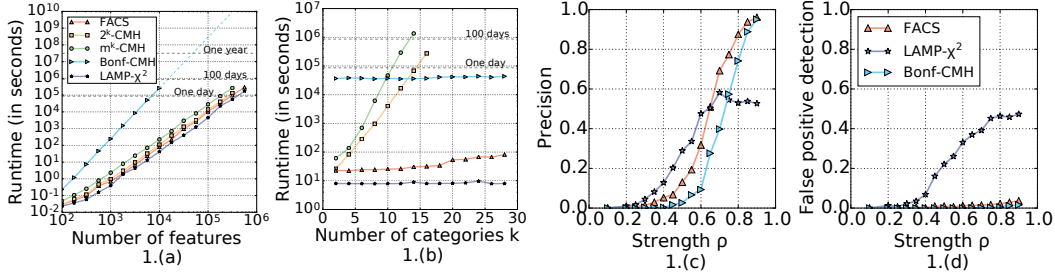

Figure 1: (a) Runtime as a function of the number of features, $p$. (b) Runtime as a function of the number of categories of the covariate, $k$. (c) Precision as a function of the true signal strengh, $\rho_{true}$. (d) False detection proportion as a function of the strength of the signal $\rho_{conf}$. $n = 200$ samples were used in (a), (b) and $n = 500$ in (c), (d). Also, we set $\rho_{true} = \rho_{conf} = \rho$.

In summary, Theorem 2 above implies that the $2^k$ candidates to be the optimum $\mathbf{x}^*_{\mathcal{S}'}$ according to Lemma 2 can be narrowed down to only $2k$ vertices: $k$ candidates satisfying the first condition and $k$ the second condition. Moreover, evaluating $\Psi_{cmh}$ for all $k$ candidates satisfying the first condition (resp. the second condition) can be done in $O(k)$ time rather than $O(k^2)$. This is due to the fact that each of the $k$ candidate vertices for each condition can be obtained by changing a single dimension with respect to the previous one. Therefore, the operation dominating the computational complexity is the sorting of the two $k$-vectors $(\beta^l_{\mathcal{S},1}, \ldots, \beta^l_{\mathcal{S},k})$ and $(\beta^r_{\mathcal{S},1}, \ldots, \beta^r_{\mathcal{S},k})$. As a consequence, the runtime required to evaluate the lower envelope $\widetilde{\Psi}_{cmh}(\mathcal{S})$, and thus our novel pruning criterion `is_prunable_cmh`, scales as $O(k \log k)$ with the number of categories of the covariate.

## 4   Experiments

In Section 4.1 we describe a set of experiments on simulated datasets, evaluating the performance of FACS in terms of runtime, precision and its ability to correct for confounding. Next, in Section 4.2, we use our method in two applications in computational biology. Due to space constraints, only a high-level summary of the experimental setup and results will be presented here. Additional details can be found in the Supp. Material and code for FACS is available on GitHub[2].

### 4.1   Runtime and power comparisons on simulated datasets

We compare FACS with four significant discriminative itemset mining methods: LAMP-$\chi^2$, Bonf-CMH, $2^k$-FACS and $m^k$-FACS. (1) LAMP-$\chi^2$ [15, 10] is the state-of-the-art in significant discriminative itemset mining. It uses Tarone's testability criterion but is based on Pearson's $\chi^2$ test and thus cannot account for covariates; (2) Bonf-CMH uses the CMH test, being able to correct for confounders, but uses Bonferroni's correction, resulting in a considerable loss of statistical power; (3) and (4) $2^k$-FACS and $m^k$-FACS are two suboptimal versions of FACS, which implement the pruning criterion using the approach shown in Lemma 2, which scales as $O(2^k)$, or via brute-force search, scaling as $O(m^k)$.

**Runtime evaluations:** Figure 1(a) shows that FACS scales as the state-of-the-art LAMP-$\chi^2$ when increasing the number of features $p$, while the Bonferroni-based method Bonf-CMH scales considerably worse. This indicates both that FACS is able to correct for covariates with virtually no runtime overhead with respect to LAMP-$\chi^2$ and confirms the efficacy of Tarone's testability criterion. Figure 1(b) shows that FACS can handle categorical covariates of high-cardinality $k$ with almost no overhead, in contrast to $m^k$-FACS and $2^k$-FACS which are only applicable for low $k$. This demonstrates the importance of our efficient implementation of the pruning criterion.

**Precision and false positive detection evaluations:** We generated synthetic datasets with one truly associated feature subset $\mathcal{S}_{true}$ and one confounded feature subset $\mathcal{S}_{conf}$ to evaluate precision and ability to correct for confounders. Figure 1(c) shows that FACS has a similar precision as LAMP-$\chi^2$, being slightly worse for weak signals and slightly better for stronger signals. Again, the performance of the Bonferroni-based method Bonf-CMH is drastically worse. Most importantly, Figure 1(d) indicates that unlike LAMP-$\chi^2$, FACS has the ability to greatly reduce the false positive detection by conditioning on an appropriate categorical covariate.

Table 1: Total number of significant combinations (hits) found by LAMP-$\chi^2$, FACS and BONF-CMH and average genomic inflation factor $\lambda$. $\lambda$ for BONF-CMH is similar to FACS since both use the CMH test.

| Datasets | FACS | | LAMP-$\chi^2$ | | BONF-CMH |
|---|---|---|---|---|---|
| | hits | $\lambda$ | hits | $\lambda$ | hits |
| LY | 433 | 1.17 | 100,883 | 3.18 | 19 |
| *avrB* | 43 | 1.21 | 546 | 2.38 | 1 |

## 4.2 Applications to computational biology

In this section, we look for significant feature combinations in two widely investigated biological applications: Genome-Wide Association Studies (GWAS), using two *A. thaliana* datasets, and a study of combinatorial regulation of gene expression in breast cancer cells.

*A. thaliana* **GWAS:** We apply FACS, LAMP-$\chi^2$ and Bonf-CMH to two datasets from the plant model organism *A. thaliana* [1], which contain 84 and 95 samples, respectively. The labels of each dataset indicate the presence/absence of a plant defense-related phenotype: LY and *avrB*. In the two datasets, each plant sample is represented by a sequence of approximately 214,000 genetic bases. The genetic bases are encoded as binary features which indicate if the base at a specific locus is standard or altered. To minimize the effect of the evolutionary correlations between nearby bases ($< 10$ kilo-bases), we downsampled each of the five chromosomes of each dataset, evenly by a factor of 20, using 20 different offsets. It resulted in complementary datasets containing between 1,423 and 2,661 features. Our results for all methods are aggregated across all downsampled versions. In GWAS, one needs to correct for the confounding effect of population structure to avoid many spurious associations. For both datasets we condition on the ancestry, resulting in $k = 5$ and $k = 3$ categories for the covariate.

Table 1 shows the number of feature combinations (c.f. Section 2.1) reported as significant by each method, as well as the corresponding genomic inflation factor $\lambda$ [4], a popular criterion in statistical genetics to quantify confounding. When compared to LAMP-$\chi^2$, we observe a severe reduction in the number of feature combinations deemed significant by FACS, as well as a sharp decrease in $\lambda$. This strongly indicates that many feature combinations reported by LAMP-$\chi^2$ are affected by confounding. The $\lambda$ values of LAMP-$\chi^2$ show strong marginal associations between many feature combinations and labels, inflating the corresponding Pearson $\chi^2$-test statistic values compared to the expected $\chi^2$ null distribution and resulting in many spurious associations. However, since most of those feature combinations are independent of the labels given the covariates, the CMH test statistics values are much closer to the $\chi^2$ distribution, leading to a lower $\lambda$ and resulting in hits that are corrected for the covariate. Moreover, the lack of power of BONF-CMH results in a very small number of hits.

**Combinatorial regulation of gene expression in breast cancer cells:** The breast cancer data set, as used in [15], includes 12,773 genes classified into up-regulated or not up-regulated. Each gene is represented by 397 binary features which indicate the presence/absence of a sequence motif in the neighborhood of this gene. We aim to find combinations of motifs that are enriched in up-regulated genes. Two sets of experiments were conducted, conditioning on 8 and 16 categories respectively. In this case, the covariate groups together genes sharing similar sets of motifs. As previously, LAMP-$\chi^2$ reports 1,214 motif combinations as significant, while FACS reports only 26 — a reduction of over 97%. Further studies shown in the Supp. Material strongly suggest that most motif combinations found by LAMP-$\chi^2$ but not FACS are indeed due to confounding.

## 5  Conclusions

This article has presented FACS, the first approach to significant discriminative itemset mining that (i) allows to condition on a categorical covariate, (ii) corrects for the inherent multiple testing problem and (iii) retains high statistical power. Furthermore, we (iv) proved that the runtime of FACS scales as $O(k \log k)$, where k is the number of states of the categorical covariate. Regarding future work, generalizing the state-of-the-art to handle continuous data is a key open problem in significant discriminative itemset mining. Solving it would greatly help make the framework applicable to new domains. Another interesting improvement would be to combine FACS with the approach in [8]. In their work, Tarone's testability criterion is used along with permutation-testing to increase statistical power by taking the redundancy between feature combinations into account. By using a similar approach in combination with the CMH test, one could further increase statistical power while retaining the ability to correct for a categorical covariate.

**Acknowledgments:**  This work was funded in part by the SNSF Starting Grant 'Significant Pattern Mining' (KB) and the Marie Curie ITN MLPM2012, Grant No. 316861 (KB, FLL).

## Footnotes

[1]We define $z_{i,\emptyset} = 1$ for all observations, so this artificial feature combination will never be significant.

[2]https://github.com/BorgwardtLab/FACS

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
