[Supplementary Material · nips16-camera-ready-supp.pdf]

# Supplementary Material for Finding significant combinations of features in the presence of categorical covariates

**Laetitia Papaxanthos**\*, **Felipe Llinares-López**\*, **Dean Bodenham, Karsten Borgwardt**
Machine Learning and Computational Biology Lab
D-BSSE, ETH Zurich

*Equally contributing authors.

## 1 Proofs of Propositions, Lemmas and Theorems

**Proposition 1** *The CMH test has a minimum attainable p-value $\Psi_{cmh}(\mathcal{S})$, which can be computed in $O(k)$ time as a function of the margins $\{n_j, n_{1,j}, x_{\mathcal{S},j}\}_{j=1}^k$ of the $k$ $2 \times 2$ contingency tables.*

**Proof:** Equation (1) in the main document can be rewritten as:

$$p_{\mathcal{S}} = 1 - F_{\chi_1^2}\left(\frac{\left(a_{\mathcal{S},tot} - \sum_{j=1}^k \frac{x_{\mathcal{S},j}n_{1,j}}{n_j}\right)^2}{\sum_{j=1}^k \frac{n_{1,j}}{n_j}\left(1 - \frac{n_{1,j}}{n_j}\right)x_{\mathcal{S},j}\left(1 - \frac{x_{\mathcal{S},j}}{n_j}\right)}\right)$$
$$= 1 - F_{\chi_1^2}(T_{\mathcal{S}}(a_{\mathcal{S},tot}, \mathbf{x}_{\mathcal{S}}))$$

where $a_{\mathcal{S},tot} = \sum_{j=1}^k a_{\mathcal{S},j}$ and $\mathbf{x}_{\mathcal{S}} = (x_{\mathcal{S},1}, \ldots, x_{\mathcal{S},k})$. Because $F_{\chi_1^2}(\cdot)$ is a monotonically increasing function of its argument $T_{\mathcal{S}}(a_{\mathcal{S},tot}, \mathbf{x}_{\mathcal{S}})$, $p_{\mathcal{S}}$ is minimized when $T_{\mathcal{S}}(a_{\mathcal{S},tot}, \mathbf{x}_{\mathcal{S}})$ is maximized. $T_{\mathcal{S}}(a_{\mathcal{S},tot}, \mathbf{x}_{\mathcal{S}})$ depends on $a_{\mathcal{S},tot}$ as a quadratic function with positive definite Hessian, hence, $p_{\mathcal{S}}$ is minimized as a function of $a_{\mathcal{S},tot}$ at the most extreme values $a_{\mathcal{S},tot}$ can attain. Since $a_{\mathcal{S},j} \in [\![a_{\mathcal{S},j,min}, a_{\mathcal{S},j,max}]\!] \; \forall j = 1, \ldots, k$, with $a_{\mathcal{S},j,min} = \max(0, x_{\mathcal{S},j} - n_{2,j})$ and $a_{\mathcal{S},j,max} = \min(x_{\mathcal{S},j}, n_{1,j})$, we have $a_{\mathcal{S},min} \le a_{\mathcal{S},tot} \le a_{\mathcal{S},max}$, where $a_{\mathcal{S},min} = \sum_{j=1}^k a_{\mathcal{S},j,min}$ and $a_{\mathcal{S},max} = \sum_{j=1}^k a_{\mathcal{S},j,max}$. Thus:

$$\Psi_{cmh}(\mathcal{S}) = 1 - F_{\chi_1^2}\left(T_{\mathcal{S}}^{max}(\mathbf{x_S})\right) \qquad (1)$$

with $T_{\mathcal{S}}^{max}(\mathbf{x_S}) = \max\left(T_{\mathcal{S}}\left(a_{\mathcal{S},min}, \mathbf{x}_{\mathcal{S}}\right), T_{\mathcal{S}}\left(a_{\mathcal{S},max}, \mathbf{x}_{\mathcal{S}}\right)\right)$ satisfies $p_{\mathcal{S}} \ge \Psi_{cmh}(\mathcal{S})$. Also, $\Psi_{cmh}(\mathcal{S})$ as defined above depends only on $\{n_j, n_{1,j}, x_{\mathcal{S},j}\}_{j=1}^k$ and can be evaluated in $O(k)$ time, which completes the proof. $\square$

**Lemma 1** *Let $\mathcal{S}, \mathcal{S}' \in \mathcal{I}_{PP}$ be two potentially prunable feature subsets such that $\mathcal{S} \subseteq \mathcal{S}'$. Then, $\widetilde{\Psi}_{cmh}(\mathcal{S}) \le \widetilde{\Psi}_{cmh}(\mathcal{S}')$ holds.*

**Proof:** The statement follows directly from the definition of the lower envelope for the CMH minimum attainable $p$-value. We have $\widetilde{\Psi}_{cmh}(\mathcal{S}) = \min_{\mathcal{S}'' \supseteq \mathcal{S}} \Psi_{cmh}(\mathcal{S}'')$ and $\widetilde{\Psi}_{cmh}(\mathcal{S}') = \min_{\mathcal{S}'' \supseteq \mathcal{S}'} \Psi_{cmh}(\mathcal{S}'')$, respectively. Also, $\mathcal{S}'' \supseteq \mathcal{S}' \Rightarrow \mathcal{S}'' \supseteq \mathcal{S}$. Thus, the set of feature subsets over which $\Psi_{cmh}(\mathcal{S}'')$ is minimized to compute $\widetilde{\Psi}_{cmh}(\mathcal{S}')$ is a subset of the set of feature subsets over which $\Psi_{cmh}(\mathcal{S}'')$ is minimized to compute $\widetilde{\Psi}_{cmh}(\mathcal{S})$. $\square$

**Theorem 1** *Let $\mathcal{S} \in \mathcal{I}_{PP}$ be a potentially prunable feature subset such that $\widetilde{\Psi}_{cmh}(\mathcal{S}) > \delta$. Then, $\Psi_{cmh}(\mathcal{S}') > \delta$ for all $\mathcal{S}' \supseteq \mathcal{S}$, i.e. all feature subsets containing $\mathcal{S}$ are non-testable at level $\delta$ and can be pruned from the search space.*

**Proof:**  Let $\mathcal{S}'$ be an arbitrary feature subset containing $\mathcal{S}$, i.e. $\mathcal{S}' \supseteq \mathcal{S}$. Then we have $\Psi_{cmh}(\mathcal{S}') \geq \widetilde{\Psi}_{cmh}(\mathcal{S}') \underset{Lemma\,1}{\geq} \widetilde{\Psi}_{cmh}(\mathcal{S})$. Therefore, $\widetilde{\Psi}_{cmh}(\mathcal{S}) > \delta \Rightarrow \Psi_{cmh}(\mathcal{S}') > \delta$. This proves that all feature subsets containing $\mathcal{S}$ are non-testable at level $\delta$. Moreover, since during the enumeration procedure described in Algorithm 2 in the main document the significance threshold $\delta$ can only decrease, those patterns can be pruned from the search space. $\square$

**Lemma 2** *Let $\mathcal{S} \in \mathcal{I}_{PP}$ be a potentially prunable feature subset. The optimum $\mathbf{x}^*_{\mathcal{S}'}$ of the discrete optimization problem $\min_{\mathbf{x}_{\mathcal{S}'} \leq \mathbf{x}_{\mathcal{S}}} \Psi_{cmh}(\mathbf{x}_{\mathcal{S}'})$ satisfies $x^*_{\mathcal{S}',j} = 0$ or $x^*_{\mathcal{S}',j} = x_{\mathcal{S},j}$ for each $j = 1, \ldots, k$*

**Proof:**  From the proof of Proposition 1 above, we have $\Psi_{cmh}(\mathcal{S}) = 1 - F_{\chi_1^2}\left(T_{\mathcal{S}}^{max}(\mathbf{x_S})\right)$ with $T_{\mathcal{S}}^{max}(\mathbf{x_S}) = \max\left(T_{\mathcal{S}}\left(a_{\mathcal{S},min}, \mathbf{x_S}\right), T_{\mathcal{S}}\left(a_{\mathcal{S},max}, \mathbf{x_S}\right)\right)$. Since $\mathcal{S} \in \mathcal{I}_{PP}$, we can write:

$$T_l(\mathbf{x_S}) := T_{\mathcal{S}}\left(a_{\mathcal{S},min}, \mathbf{x_S}\right) = \frac{\left(\sum_{j=1}^{k} \gamma_j x_{\mathcal{S},j}\right)^2}{\sum_{j=1}^{k} \gamma_j(1-\gamma_j)x_{\mathcal{S},j}\left(1 - \frac{x_{\mathcal{S},j}}{n_j}\right)} \tag{2}$$

$$T_r(\mathbf{x_S}) := T_{\mathcal{S}}\left(a_{\mathcal{S},max}, \mathbf{x_S}\right) = \frac{\left(\sum_{j=1}^{k} (1-\gamma_j) x_{\mathcal{S},j}\right)^2}{\sum_{j=1}^{k} \gamma_j(1-\gamma_j)x_{\mathcal{S},j}\left(1 - \frac{x_{\mathcal{S},j}}{n_j}\right)} \tag{3}$$

where we have used that $\mathcal{S} \in \mathcal{I}_{PP} \Rightarrow x_{\mathcal{S},j} \leq \min(n_{1,j}, n_{2,j}) \; \forall\, j = 1, \ldots, k$ and defined the class ratios $\gamma_j := \min(n_{1,j}, n_{2,j})/n_j$ for each $j = 1, \ldots, k$. Note also that minimising $\Psi_{cmh}(\mathbf{x}_{\mathcal{S}'})$ on $\mathbf{x}_{\mathcal{S}'} \leq \mathbf{x}_{\mathcal{S}}$ is in this case equivalent to maximising the maximum between $T_l(\mathbf{x}_{\mathcal{S}'})$ and $T_r(\mathbf{x}_{\mathcal{S}'})$.

As a first step towards proving Lemma 2, we will show that the functions $T_l(\mathbf{x}_{\mathcal{S}'})$ and $T_r(\mathbf{x}_{\mathcal{S}'})$ are both maximised with respect to a single argument $x_{\mathcal{S}',i}$ while keeping the other arguments $x_{\mathcal{S}',j}, j \neq i$ fixed at either: (I) $x_{\mathcal{S}',i} = 0$ or (II) $x_{\mathcal{S}',i} = x_{\mathcal{S},i}$. To show that, we compute the partial derivative of $T_l(\mathbf{x}_{\mathcal{S}'})$ and $T_r(\mathbf{x}_{\mathcal{S}'})$ with respect to $x_{\mathcal{S}',i}$:

$$\frac{\partial T_l(\mathbf{x}_{\mathcal{S}'})}{\partial x_{\mathcal{S}',i}} = \Lambda_l(\mathbf{x}_{\mathcal{S}'})A_l(\mathbf{x}_{\mathcal{S}'})$$

$$\frac{\partial T_r(\mathbf{x}_{\mathcal{S}'})}{\partial x_{\mathcal{S}',i}} = \Lambda_r(\mathbf{x}_{\mathcal{S}'})A_r(\mathbf{x}_{\mathcal{S}'})$$

with

$$\Lambda_l(\mathbf{x}_{\mathcal{S}'}) = \frac{\gamma_i \sum_{j=1}^{k} \gamma_j x_{\mathcal{S}',j}}{\left(\sum_{j=1}^{k} \gamma_j(1-\gamma_j)x_{\mathcal{S}',j}(1-\frac{x_{\mathcal{S}',j}}{n_j})\right)^2}$$

$$\Lambda_r(\mathbf{x}_{\mathcal{S}'}) = \frac{(1-\gamma_i) \sum_{j=1}^{k} (1-\gamma_j) x_{\mathcal{S}',j}}{\left(\sum_{j=1}^{k} \gamma_j(1-\gamma_j)x_{\mathcal{S}',j}(1-\frac{x_{\mathcal{S}',j}}{n_j})\right)^2}$$

$$A_l(\mathbf{x}_{\mathcal{S}'}) = \sum_{j=1,j\neq i}^{k} \gamma_j x_{\mathcal{S}',j}\left(2(1-\gamma_j)(1-\frac{x_{\mathcal{S}',j}}{n_j}) - (1-\gamma_i)\right) + (1-\gamma_i)\left(\gamma_i + \frac{2}{n_i}\sum_{j=1,j\neq i}^{k} \gamma_j x_{\mathcal{S}',j}\right)x_{\mathcal{S}',i}$$

$$A_r(\mathbf{x}_{\mathcal{S}'}) = \sum_{j=1,j\neq i}^{k} (1-\gamma_j) x_{\mathcal{S}',j}\left(2\gamma_j(1-\frac{x_{\mathcal{S}',j}}{n_j}) - \gamma_i\right) + \gamma_i\left((1-\gamma_i) + \frac{2}{n_i}\sum_{j=1,j\neq i}^{k} (1-\gamma_j) x_{\mathcal{S}',j}\right)x_{\mathcal{S}',i}$$

Because $\Lambda_l(\mathbf{x}_{\mathcal{S}'}) \geq 0$ and $\Lambda_r(\mathbf{x}_{\mathcal{S}'}) \geq 0$, the sign of the partial derivatives are determined by the sign of $A_l(\mathbf{x}_{\mathcal{S}'})$ and $A_r(\mathbf{x}_{\mathcal{S}'})$ respectively. In both cases, $A(\mathbf{x}_{\mathcal{S}'})$ can be expressed as $A(\mathbf{x}_{\mathcal{S}'}) =$

$b(\mathbf{x}_{\neg i, \mathcal{S}'}) + \mu(\mathbf{x}_{\neg i, \mathcal{S}'})x_{\mathcal{S}',i}$, with $\mathbf{x}_{\neg i, \mathcal{S}'}$ containing $\{x_{\mathcal{S}',j}\}_{j=1, j\neq i}^{k}$. That is, $A(\mathbf{x}_{\mathcal{S}'})$ is an affine function of $x_{\mathcal{S}',i}$ where the intersect and slope is controlled by all other $k-1$ variables. Moreover, for any $\mathbf{x}_{\neg i, \mathcal{S}'}$ we have $\mu(\mathbf{x}_{\neg i, \mathcal{S}'}) \geq 0$. Therefore the partial derivatives are either always positive, always negative, or negative until a unique point where it crosses zero and then positive. As a consequence, it follows that the only possible maximizers of $T_l(\mathbf{x}_{\mathcal{S}'})$ and $T_r(\mathbf{x}_{\mathcal{S}'})$ with respect to $x_{\mathcal{S}',i}$ are at the boundary of the domain, i.e. either $x_{\mathcal{S}',i} = 0$ or $x_{\mathcal{S}',i} = x_{\mathcal{S},i}$. In other words, we have

$$\max_{\mathbf{x}_{\mathcal{S}'} \leq \mathbf{x}_{\mathcal{S}}} T_l(\mathbf{x}_{\mathcal{S}'}) = \max\left(\max_{\mathbf{x}_{\neg i, \mathcal{S}'} \leq \mathbf{x}_{\neg i, \mathcal{S}}} T_l(\mathbf{x}_{\neg i, \mathcal{S}'}, 0), \max_{\mathbf{x}_{\neg i, \mathcal{S}'} \leq \mathbf{x}_{\neg i, \mathcal{S}}} T_l(\mathbf{x}_{\neg i, \mathcal{S}'}, x_{\mathcal{S},i})\right).$$ Since this holds

for any $i = 1, 2, \ldots, k$ and $\mathbf{x}_{\mathcal{S}'} \leq \mathbf{x}_{\mathcal{S}}$, the argument can be applied recursively to each of the two terms in the RHS of the last expression. This completes the proof. $\qquad\square$

**Theorem 2** *Let $\mathcal{S} \in \mathcal{I}_{PP}$ be a potentially testable feature subset and define $\beta_{\mathcal{S},j}^{l} = \frac{n_{2,j}}{n_j}\left(1 - \frac{x_{\mathcal{S},j}}{n_j}\right)$ and $\beta_{\mathcal{S},j}^{r} = \frac{n_{1,j}}{n_j}\left(1 - \frac{x_{\mathcal{S},j}}{n_j}\right)$ for $j = 1, \ldots, k$. Let $\pi_l$ and $\pi_r$ be permutations $\pi_l, \pi_r : [\![1, k]\!] \mapsto [\![1, k]\!]$ such that $\beta_{\mathcal{S},\pi_l(1)}^{l} \leq \cdots \leq \beta_{\mathcal{S},\pi_l(k)}^{l}$ and $\beta_{\mathcal{S},\pi_r(1)}^{r} \leq \cdots \leq \beta_{\mathcal{S},\pi_r(k)}^{r}$, respectively.*

*Then, there exists an integer $\kappa \in [\![1, k]\!]$ such that the optimum $\mathbf{x}_{\mathcal{S}'}^{*} = \arg\min_{\mathbf{x}_{\mathcal{S}'} \leq \mathbf{x}_{\mathcal{S}}} \Psi_{cmh}(\mathbf{x}_{\mathcal{S}'})$ satisfies one of the two possible conditions: (I) $x_{\mathcal{S}',\pi_l(j)}^{*} = x_{\mathcal{S},\pi_l(j)}$ for all $j \leq \kappa$ and $x_{\mathcal{S}',\pi_l(j)}^{*} = 0$ for all $j > \kappa$ or (II) $x_{\mathcal{S}',\pi_r(j)}^{*} = x_{\mathcal{S},\pi_r(j)}$ for all $j \leq \kappa$ and $x_{\mathcal{S}',\pi_r(j)}^{*} = 0$ for all $j > \kappa$.*

**Proof:** The functions $T_l(\mathbf{x}_{\mathcal{S}'})$ and $T_r(\mathbf{x}_{\mathcal{S}'})$ defined in the proof of Lemma 2 above can be rewritten generically as:

$$T = \frac{\left(\sum_{j=1}^{k} l_j(\beta_j)\right)^2}{\sum_{j=1}^{k} \beta_j l_j(\beta_j)} \tag{4}$$

with $\beta_j \in [0, 1]$ and $l_j(\beta_j) > 0$. In particular, $\beta_j = (1 - \gamma_j)(1 - \frac{x_{\mathcal{S},j}}{n_j})$ and $l_j(\beta_j) = n_{1,j}\left(1 - \frac{\beta_j}{1 - \gamma_j}\right)$ for $T_l(\mathbf{x}_{\mathcal{S}'})$ whereas $\beta_j = \gamma_j(1 - \frac{x_{\mathcal{S},j}}{n_j})$ and $l_j(\beta_j) = n_{2,j}\left(1 - \frac{\beta_j}{1 - \gamma_j}\right)$ for $T_r(\mathbf{x}_{\mathcal{S}'})$. Since $T$ is permutation invariant, we assume without loss of generality that the indices $j = 1, \ldots, k$ have been sorted a priori to guarantee that $\beta_j \leq \beta_i$ whenever $j \leq i$.

Next, we introduce $k$ binary indicator variables $\delta_1, \ldots, \delta_k$ in $T$ as:

$$T(\delta_1, \ldots, \delta_k) = \frac{\left(\sum_{j=1}^{k} \delta_j l_j(\beta_j)\right)^2}{\sum_{j=1}^{k} \delta_j \beta_j l_j(\beta_j)} \tag{5}$$

Suppose further that:

$$\arg\max_{\delta_1, \ldots, \delta_k} T(\delta_1, \ldots, \delta_k) = (\underbrace{1, 1, \ldots, 1}_{r}, \underbrace{0, 0, \ldots, 0}_{k-r}) \tag{6}$$

holds with $r > 0$. Informally, Equation (6) being true would imply that the maximum is achieved by keeping the terms in the summation corresponding to the $r$ smallest $\beta_j$. From Lemma 2, we know that $T_l(\mathbf{x}_{\mathcal{S}'})$ and $T_r(\mathbf{x}_{\mathcal{S}'})$ are maximised for $x_{\mathcal{S}',j}^{*} = 0$ or $x_{\mathcal{S}',j}^{*} = x_{\mathcal{S},j}$ for all $j = 1, 2, \ldots, k$. Thus, if Equation (6) holds and we have $\beta_i > \beta_j$ and $x_{\mathcal{S}',i}^{*} = x_{\mathcal{S},i}$ then it follows that $x_{\mathcal{S}',j}^{*} = x_{\mathcal{S},j}$. The alternative case $x_{\mathcal{S}',j}^{*} = 0$ cannot occur, since it would contradict Equation (6). This would suggest the following strategy to solve $\min_{\mathbf{x}_{\mathcal{S}'} \leq \mathbf{x}_{\mathcal{S}}} \Psi_{cmh}(\mathbf{x}_{\mathcal{S}'})$. Firstly, as shown in the proof of Lemma 2, $\arg\min_{\mathbf{x}_{\mathcal{S}'} \leq \mathbf{x}_{\mathcal{S}}} \Psi_{cmh}(\mathbf{x}_{\mathcal{S}'}) = \arg\max_{\mathbf{x}_{\mathcal{S}'} \leq \mathbf{x}_{\mathcal{S}}} \max(T_l(\mathbf{x}_{\mathcal{S}'}), T_r(\mathbf{x}_{\mathcal{S}'}))$. Next, we obtain and sort the coefficients $\{\beta_j^{l}\}_{j=1}^{k}$ and $\{\beta_j^{r}\}_{j=1}^{k}$ corresponding to the representation of $T_l$ and $T_r$ in the form of Equation (4). The computational complexity of that step would be dominated by the sorting steps, hence being $O(k \log(k))$. Then, by Equation (6) and Lemma 2, we can solve the subproblems $\arg\max_{\mathbf{x}_{\mathcal{S}'} \leq \mathbf{x}_{\mathcal{S}}} T_l(\mathbf{x}_{\mathcal{S}'})$ and $\arg\max_{\mathbf{x}_{\mathcal{S}'} \leq \mathbf{x}_{\mathcal{S}}} T_r(\mathbf{x}_{\mathcal{S}'})$ in $O(k)$ time each, increasing the candidate $r$ in Equation (6) from 1 up to at

most $k$. Note that this is exactly the strategy suggested by Theorem 2. In summary, proving Theorem 2 amounts to showing the validity of Equation (6) for functions of the form given in Equation (5).

We will prove it by induction. First, we show that the statement holds for $k = 2$. That is, we want to show that:

$$\arg\max_{\delta_1,\delta_2} T(\delta_1, \delta_2) \in \{(1,0), (1,1)\} \tag{7}$$

The only possible contradicting case would be $\arg\max_{\delta_1,\delta_2} T(\delta_1, \delta_2) = (0,1)$, since the case $(0,0)$ yields a trivial value for the function T. We show directly that under the assumption $\beta_1 \leq \beta_2$, the contradiction cannot happen. Indeed we have:

$$\frac{(l_1(\beta_1) + l_2(\beta_2))^2}{\beta_1 l_1(\beta_1) + \beta_2 l_2(\beta_2)} - \frac{l_2^2(\beta_2)}{\beta_2 l_2(\beta_2)} = \frac{(l_1(\beta_1) + l_2(\beta_2))^2 \beta_2 l_2(\beta_2) - l_2^2(\beta_2)(\beta_1 l_1(\beta_1) + \beta_2 l_2(\beta_2))}{(\beta_1 l_1(\beta_1) + \beta_2 l_2(\beta_2))\beta_2 l_2(\beta_2)}$$

$$\tag{8}$$

$$= l_1(\beta_1) \frac{(l_1(\beta_1) + 2l_2(\beta_2))\beta_2 l_2(\beta_2) - \beta_1 l_2^2(\beta_2)}{(\beta_1 l_1(\beta_1) + \beta_2 l_2(\beta_2))\beta_2 l_2(\beta_2)}$$

$$= l_1(\beta_1) \frac{\beta_2 l_1(\beta_1) l_2(\beta_2) + (2\beta_2 - \beta_1)l_2^2(\beta_2)}{(\beta_1 l_1(\beta_1) + \beta_2 l_2(\beta_2))\beta_2 l_2(\beta_2)}$$

Since $l_i(\beta_i) \geq 0$ and $\beta_1 \leq \beta_2$, it follows that the numerator in the expression above is positive, thus $T(1,1) > T(0,1)$ contradicting the statement that $\arg\max_{\delta_1,\delta_2} T(\delta_1, \delta_2) = (0,1)$.

Next we prove the induction step. Suppose the statement holds for an arbitrary dimension $k$, we will show then it also holds for dimension $k + 1$. That is, if we have:

$$\arg\max_{\delta_1,\ldots,\delta_k} \frac{\left(\sum_{i=1}^{k} \delta_i l_i(\beta_i)\right)^2}{\sum_{i=1}^{k} \delta_i \beta_i l_i(\beta_i)} = (\underbrace{1,1,\ldots,1}_{R}, \underbrace{0,0,\ldots,0}_{k-R}) \tag{9}$$

Then we want to show that:

$$\arg\max_{\delta_1,\ldots,\delta_k,\delta_{k+1}} \frac{\left(\sum_{i=1}^{k} \delta_i l_i(\beta_i) + \delta_{k+1} l_{k+1}(\beta_{k+1})\right)^2}{\sum_{i=1}^{k} \delta_i \beta_i l_i(\beta_i) + \delta_{k+1} \beta_{k+1} l_{k+1}(\beta_{k+1})} = (\underbrace{1,1,\ldots,1}_{R'}, \underbrace{0,0,\ldots,0}_{(k+1)-R'}) \tag{10}$$

We can start by writing:

$$\max_{\delta_1,\ldots,\delta_k,\delta_{k+1}} \frac{\left(\sum_{i=1}^{k} \delta_i l_i(\beta_i) + \delta_{k+1} l_{k+1}(\beta_{k+1})\right)^2}{\sum_{i=1}^{k} \delta_i \beta_i l_i(\beta_i) + \delta_{k+1} \beta_{k+1} l_{k+1}(\beta_{k+1})} \tag{11}$$

$$= \max\left(\max_{\delta_1,\ldots,\delta_k} \frac{\left(\sum_{i=1}^{k} \delta_i l_i(\beta_i)\right)^2}{\sum_{i=1}^{k} \delta_i \beta_i l_i(\beta_i)}, \max_{\delta_1,\ldots,\delta_k} \frac{\left(\sum_{i=1}^{k} \delta_i l_i(\beta_i) + l_{k+1}(\beta_{k+1})\right)^2}{\sum_{i=1}^{k} \delta_i \beta_i l_i(\beta_i) + \beta_{k+1} l_{k+1}(\beta_{k+1})}\right)$$

If:

$$\max_{\delta_1,\ldots,\delta_k} \frac{\left(\sum_{i=1}^{k} \delta_i l_i(\beta_i)\right)^2}{\sum_{i=1}^{k} \delta_i \beta_i l_i(\beta_i)} \geq \max_{\delta_1,\ldots,\delta_k} \frac{\left(\sum_{i=1}^{k} \delta_i l_i(\beta_i) + l_{k+1}(\beta_{k+1})\right)^2}{\sum_{i=1}^{k} \delta_i \beta_i l_i(\beta_i) + \beta_{k+1} l_{k+1}(\beta_{k+1})} \tag{12}$$

Then the statement is trivially true. Suppose now that equation (12) does **not** hold. We show next that:

$$(\hat{\delta}_1, \ldots, \hat{\delta}_k) = \arg\max_{\delta_1,\ldots,\delta_k} \frac{\left(\sum_{i=1}^{k} \delta_i l_i(\beta_i) + l_{k+1}(\beta_{k+1})\right)^2}{\sum_{i=1}^{k} \delta_i \beta_i l_i(\beta_i) + \beta_{k+1} l_{k+1}(\beta_{k+1})} = (\underbrace{1,1,\ldots,1}_{k}) \tag{13}$$

which would complete the proof. To show that equation (13) is true when equation (12) does **not** hold, we proceed by contradiction in two steps. First we prove that there is at most a single $j \in \{1, \ldots, k\} \mid \hat{\delta}_j = 0$. To see that, suppose $\exists j \mid \hat{\delta}_j = 0$ and define:

$$\widetilde{T}(\delta_1, \ldots, \delta_{j-1}, \delta_{j+1}, \ldots, \delta_k, \delta_{k+1}) = \frac{\left(\sum_{i=1, i \neq j}^{k} \delta_i l_i(\beta_i) + \delta_{k+1} l_{k+1}(\beta_{k+1})\right)^2}{\sum_{i=1, i \neq j}^{k} \delta_i \beta_i l_i(\beta_i) + \delta_{k+1} \beta_{k+1} l_{k+1}(\beta_{k+1})} \quad (14)$$

which is nothing but $T(\delta_1, \delta_2, \ldots, \delta_k, \delta_{k+1})$ with the $j$-th term removed. Note that, since $\hat{\delta}_j = 0$, we have:

$$\max_{\delta_1, \ldots, \delta_{j-1}, \delta_{j+1}, \ldots, \delta_k, \delta_{k+1}} \widetilde{T}(\delta_1, \ldots, \delta_{j-1}, \delta_{j+1}, \ldots, \delta_k, \delta_{k+1}) \geq \max_{\delta_1, \ldots, \delta_k} \frac{\left(\sum_{i=1}^{k} \delta_i l_i(\beta_i) + l_{k+1}(\beta_{k+1})\right)^2}{\sum_{i=1}^{k} \delta_i \beta_i l_i(\beta_i) + \beta_{k+1} l_{k+1}(\beta_{k+1})} \quad (15)$$

Moreover, since equation (12) does not hold, it follows that:

$$(\tilde{\delta}_1, \ldots, \tilde{\delta}_{j-1}, \tilde{\delta}_{j+1}, \ldots, \tilde{\delta}_k, \tilde{\delta}_{k+1}) = \underset{\delta_1, \ldots, \delta_{j-1}, \delta_{j+1}, \ldots, \delta_k, \delta_{k+1}}{\arg\max} \widetilde{T}(\delta_1, \ldots, \delta_{j-1}, \delta_{j+1}, \ldots, \delta_k, \delta_{k+1}) \quad (16)$$

satisfies $\tilde{\delta}_{k+1} = 1$ (otherwise equation (12) would be true). But, since the monotonicity property is assumed to be true for problems of dimension $k$, it turns out that $\tilde{\delta}_i = 1$ for $i = 1, \ldots, j-1, j+1, \ldots, k$ as well. And, since $\tilde{\delta}_{k+1} = 1$, then:

$$\max_{\delta_1, \ldots, \delta_{j-1}, \delta_{j+1}, \ldots, \delta_k, \delta_{k+1}} \widetilde{T}(\delta_1, \ldots, \delta_{j-1}, \delta_{j+1}, \ldots, \delta_k, \delta_{k+1}) = \max_{\delta_1, \ldots, \delta_k} \frac{\left(\sum_{i=1}^{k} \delta_i l_i(\beta_i) + l_{k+1}(\beta_{k+1})\right)^2}{\sum_{i=1}^{k} \delta_i \beta_i l_i(\beta_i) + \beta_{k+1} l_{k+1}(\beta_{k+1})} \quad (17)$$

which in turn implies $\hat{\delta}_i = 1$ for $i = 1, \ldots, j-1, j+1, \ldots, k$. Thus, $j$ is the only dimension which could satisfy $\hat{\delta}_j = 0$.

To end the proof, we need to show that, indeed, it's not possible to have $\hat{\delta}_j = 0$ either. To do so we will show that the statement of monotonicity holds for $k = 3$, then we could easily show $\hat{\delta}_j = 1$. We use a change of variables to make it clearer.

Indeed, we rewrite $T(\delta_1, \delta_2, ..., \delta_j, ..., \delta_{k-1}, \delta_k)$ the following way :

$$T(\delta_1, \delta_2, ..., \delta_{k-1}, \delta_k) = \frac{(f_0 + f_j \delta_i + f_k)^2}{\beta_0 f_0 + \beta_j f_j \delta_i + \beta_k f_k}$$

with

$$f_0 = \sum_{l=1...k-1/j} f_l$$

and

$$f_0 \beta_0 = \sum_{l=1...k-1/j} f_l \beta_l \Leftrightarrow \beta_0 = \frac{\sum_{l=1...k-1/j} f_l \beta_l}{\sum_{l=1...k-1/j} f_l}$$

Then, if equation (12) does **not** hold, we would have:

$$\max_{\delta_1, \ldots, \delta_k, \delta_{k+1}} T(\delta_1, \ldots, \delta_k, \delta_{k+1}) = \max_{\delta_0, \delta_j, \delta_{k+1}} \frac{(\delta_0 l_0 + \delta_j l_j + \delta_{k+1} l_{k+1})^2}{\delta_0 \beta_0 l_0 + \delta_j \beta_j l_j + \delta_{k+1} \beta_{k+1} l_{k+1}} \quad (18)$$

where we know, by assumption, that the optimum in the right hand side is achieved when $\delta_0 = 1$ and $\delta_{k+1} = 1$. If we knew monotonicity holds for k=3, it would then follow that $\delta_j = 1$ if $\beta_j \geq \beta_0$.

To rephrase it, we want to show that the two following cases: $T(\delta_j = 1, \delta_0 = 1, \delta_k = 1) < T(\delta_j = 0, \delta_0 = 1, \delta_k = 1)$ with $\beta_j < \beta_0$ and $T(\delta_j = 1, \delta_0 = 1, \delta_k = 1) < T(\delta_j = 0, \delta_0 = 1, \delta_k = 1)$

with $\beta_j > \beta_0$ are impossible with the hypothesis that $\forall \quad \{\delta_1, \delta_2, ..., \delta_{k-1}\} \quad T(\delta_1, \delta_2, ..., \delta_{k-1}, 0) < max_{\delta_1,...,\delta_{k-1}} T(\delta_1, \delta_2, ..., \delta_{k-1}, 1)$

First, we show that when $\beta_j < \beta_0$, then $T(\delta_j = 1, \delta_0 = 1, \delta_k = 1) > T(\delta_j = 0, \delta_0 = 1, \delta_k = 1)$. Indeed, after developing the difference we obtain :

$$T(\delta_j = 1, \delta_0 = 1, \delta_k = 1) - T(\delta_j = 0, \delta_0 = 1, \delta_k = 1) \tag{19}$$

$$= \frac{l_j}{(l_0\beta_0 + l_k\beta_k)(l_j\beta_j + l_0\beta_0 + l_k\beta_k)}(l_0^2(2\beta_0 - \beta_j) \tag{20}$$

$$+ l_k^2(2\beta_k - \beta_i) + l_0 l_k(2\beta_0 + 2\beta_k - \beta_j + l_0 l_j \beta_0 + l_k l_j \beta_k)) \qquad > 0 \tag{21}$$

As $\beta_j < \beta_0 < \beta_k$, all the terms of the previous sum are positive, which implies that $T(\delta_j = 1, \delta_0 = 1, \delta_k = 1) > T(\delta_j = 0, \delta_0 = 1, \delta_k = 1)$.

In a second time we want to show that the case $T(\delta_0 = 1, \delta_j = 1, \delta_k = 1) < T(\delta_0 = 1, \delta_j = 0, \delta_k = 1)$ with $\beta_j > \beta_0$ is not possible either. In this case we use a Reductio ad absurdum: we are going to show that we can not have both $T(\delta_0 = 1, \delta_j = 0, , 1) > T(1, 1, 1)$ and $T(\delta_0 = 1, \delta_j = 0, 1) > T(\delta_0 = 0, \delta_j = 1, 0)$. Indeed after developing both inequalities, we find

$$T(\delta_0 = 1, \delta_j = 0, 1) > T(1, 1, 1) \Leftrightarrow \beta_0 < \frac{1}{l_0}(\beta_j \frac{(l_0 + l_k)^2}{2(l_0 + l_k) + l_j} - \beta_k l_k) \tag{22}$$

$$T(\delta_0 = 1, \delta_j = 0, 1) > T(\delta_0 = 1, \delta_j = 0, 0) \Leftrightarrow \beta_0 > \frac{l_0}{2l_0 + l_k}\beta_k \tag{23}$$

The first inequality of 22 can be simplified the following way, by using the following inequalities $\beta_0 < \beta_j < \beta_k$ and $\forall i \quad l_i > 0$.

$$\beta_0 < \frac{1}{l_0}(\beta_i \frac{(l_0 + l_k)^2}{2(l_0 + l_k) + l_j} - \beta_k l_k) \tag{24}$$

$$< \frac{1}{l_0}(\beta_k \frac{(l_0 + l_k)^2}{2(l_0 + l_k) + l_j} - \beta_k l_k) = \beta_k(\frac{1}{l_0}\frac{(l_0 + l_k)^2}{2(l_0 + l_k) + l_j} - l_k) \tag{25}$$

Using 22 and 24 we have the following result :

$$\frac{l_0}{2l_0 + l_k}\beta_k < \beta_0 < \beta_k \frac{1}{l_0}(\frac{(l_0 + l_k)^2}{2(l_0 + l_k) + l_j} - l_k) \tag{26}$$

$$\Rightarrow \frac{l_0}{2l_0 + l_k} < \frac{1}{l_0}(\frac{(l_0 + l_k)^2}{2(l_0 + l_k) + l_i} - l_k) \tag{27}$$

$$\Rightarrow 0 < -(l_0 + l_k)^2(l_k + l_j) \tag{28}$$

The last line of the previous equation set shows clearly the contradiction.

Those two results 19 and 26 end the proof.

$\square$

# 2 Experimental results

## 2.1 Implementation details

All algorithms considered in the experiments used the same closed itemset mining algorithm for enumerating patterns: a simplified version of the Eclat algorithm based on an implementation by the author in [4]. All approaches were implemented using C++ and compiled with the same flags.

In this way, the runtime differences among methods due to implementation, rather than algorithmic considerations, should be minimal.

All experiments for which runtime values are reported were carried out using a single thread running on a Intel Xeon E5-2680v3 CPU, clocked at 2.5 GHz. While the maximum amount of RAM available in the system was 64 GB, less than 8 GB were needed throughout the experiments.

## 2.2 Simulation experiments

### 2.2.1 Data generation

We generate synthetic transaction databases containing $n$ observations $\{\mathbf{u}_i\}_{i=1}^n$ and $p$ binary features each. Each feature $u_{i,j} \in \{1, \ldots, p\}$ takes the value 1 in each observation $\{u_i\}_{i=1}^n$ according to the realization of a Bernoulli random variable with parameter $\mu$. Different realizations are i.i.d. across observations and features. In addition, each of the observations $\mathbf{u}_i$ was assigned a binary class label $y_i$ and a categorical covariate $c_i \in \{1, \ldots, k\}$.

A true associated feature combination $\mathbf{z}_{true}$ was generated as a binary vector correlated to the label-vector $\mathbf{y}$ but not correlated to the covariate-vector $\mathbf{c}$. In addition, a confounded feature combination $\mathbf{z}_{conf}$ was generated as a binary vector almost fully correlated to the covariate-vector $\mathbf{c}$. Those two feature combinations were further decomposed into 5 feature vectors such that the AND operation of all those individual feature vectors gives the respective combination features. Those 10 feature vectors replace 10 observations chosen at random in the generated dataset. The strength of the correlations between the true associated feature combinations is controlled by $\rho_{true}$. $\rho_{conf}$ strictly controls the association between the class label and the covariate $\mathbf{c}$, and as a consequence the association between the confounded itemset and the class label, because their correlation almost reaches one (c.f. the following paragraph). As an example, those parameters set to one indicate a perfect match.

**Generating distribution** We now describe a procedure for generating a *confounded significant feature combination*. Consider the covariance matrix

$$\Sigma = \begin{pmatrix} 1 & 0 & \rho_{\text{true}} \\ 0 & 1 & \rho_{\text{conf}} \\ \rho_{\text{true}} & \rho_{\text{conf}} & 1 \end{pmatrix}$$

Consider sampling a single drawing $o$ from the multivariate Bernoulli distribution with mean $(0.5, 0.5, 0.5)$ and covariance matrix $\Sigma$. This can be done in R using the *bindata* package, which results in a three-vector $o = (u, c, y)$.

This $o$ will be three-dimensional; the first component $o_1$ is an indicator function, which indicates (1) if that feature $u_i$ will contain the studied combination for sample $i$, or not (0). The second component $o_2$ is the categorical label, and the third component $z_3$ will be the class label $y_i$. In this way we generate the class vector $\mathbf{c}$ and the true associated significant itemset $\mathbf{u_{true}}$.

For the confounded itemsets, each feature combination $\mathbf{z}_{conf,i}$ is obtained from the values of the categorical covariate $c_i$ by flipping its value with a low probability $p_\epsilon = 0.05$. To be clear, we sample $\epsilon \sim \mathrm{B}(1, p_\epsilon)$ and then

$$z_{conf,i} = c_i \oplus \epsilon,$$

where $\oplus$ is the **xor** operator. By looking at $\Sigma$, we see that the parameter $\rho_{conf}$ controls the degree of association between $c$ and $y$. For high values of $\rho_{\text{con}}$, the vectors $\mathbf{c}$ and $\mathbf{z_{conf}}$ will be highly correlated with $\mathbf{y}$.

Let $p \sim \mathrm{B}(\mu)$, $f$ written such that $f = \mathrm{AND}(o_2, p)$ is partly a confounded itemset, because its subset contains a confounded feature.

### 2.2.2 Runtime evaluation

We evaluated the speed of our method while varying two fundamental parameters: the number of features $p$ and the number of categories for the covariate $k$, while keeping $n = 500$ and $\mu = 0.1$.

Figure 1.(a) (c.f. main text) shows the runtime as a function of the number of items, $p$. This is a fundamental parameter, as databases in the applications we target, such as computational biology, are often characterized by a small sample size $n$ and a large number of input features $p$. The main

observation one can derive from Figure 1.(a) is that, again, `FACS` is virtually as fast as state-of-the-art unconditional contrast pattern mining. We can also notice that methods using Tarone's testability criterion are vastly more efficient than the Bonferroni-based method `Bonf-CMH`. The difference in performance gets particularly relevant for sufficiently large $p$.

Figure 1.(b) (c.f. main text) shows the runtime as a function of the number of categories for the covariate, $k$. The runtime of `FACS` can be seen to scale slowly with $k$, as expected from the result in Theorem 2. The overhead with respect to unconditional pattern mining, represented by `LAMP`-$\chi^2$, is small even for as many as $k = 26$ different categories for the covariate. In contrast, the runtime of $m^k$-`FACS`, which uses a naive-implementation of the pruning criterion, and $2^k$-`FACS`, which uses a suboptimal implementation based on Lemma 2, increases exponentially with $k$. In summary, this experiment demonstrates that `FACS` can scale to large values of $k$ with only a minor overhead and shows the importance of our efficient implementation of the pruning criterion to achieve that result.

### 2.2.3   Evaluation of precision and false positive detection

In the main text, we describe the performance of `FACS` in terms of: (a) precision, defined as the proportion of itemsets deemed significant which are true positives, and (b) false positive detection, defined as the proportion itemsets deemed significant which are confounded. To do so, we generated 300 synthetic databases as described in Section 2.2.1 for different values of $\rho_{true}$ and $\rho_{conf}$. In our simulations, we assume that the strength of the association between the true itemset and the label $\rho_{true}$, and the confounded itemset and the label, $\rho_{conf}$, are identical, i.e. that $\rho_{true} = \rho_{conf}$. In this way, we do not favor the detection of true itemsets over confounded itemsets or vice versa. All synthetic databases were generated using $n = 200$, $p = 5000$ and $\mu = 0.1$. Each itemset reported by a method as significantly associated will be considered a *true positive detection* if strictly more than half of its items belong to the true itemset. Analogously, they will be considered a *false positive detection* if strictly more than half of its items belong to the confounded itemset. Itemsets that contain as many items belonging to the true associated feature combination as items belonging to the confounded feature combination count as half of an itemset for each of the category when summing the number of hits (true and false hits).

Figures 1.(c) in the main text compare `FACS`, `LAMP`-$\chi^2$ and Bonf-CMH. The precision of `FACS` and `LAMP`-$\chi^2$, both of which employ Tarone's testability criterion, is superior to that of `BONF-CMH`, as expected. Differences are more accentuated for moderate association strengths, which correspond to signals strong enough to be detectable yet not considerably above the noise level. More importantly, in Figure 1.(d) we observe that unconditional significant discrimative itemset mining methods such as `LAMP`-$\chi^2$ have an unacceptably high proportion of confounded items being deemed significant. In contrast, `FACS` greatly reduces the false positive detection by conditioning on an appropriate covariate. Finally, the false positive detection of `BONF-CMH` is even lower than that of `FACS`, a consequence of the low statistical power of methods based on Bonferroni's correction.

### 2.3   Experiments on biomedical datasets

As mentioned in the main text we apply our method to solve two different biological questions.

#### 2.3.1   Dataset descriptions and preprocessing

***A. thaliana* GWAS** We analyze a widely used collection of *Arabidopsis thaliana* GWAS datasets from [3], which we obtained from the easyGWAS online resource [7]. We chose two representative datasets among the ones exhibiting the highest amount of confounding, as measured by the genomic inflation factor $\lambda$ described in [6]. The two datasets chosen concern the disease LY (yellowing leaves) and *avrB* (hypersensitive-response traits). They contain 84 and 95 samples respectively and approximately 214,000 binary features. A feature takes value 1 if the corresponding position in the genomic sequence is a *minor allele*, i.e. the less frequent genomic variant in the population, and value 0 otherwise.

We consider the datasets of each plant trait, LY or *avrB*, and downsample the two datasets into smaller datasets: (1) according to the chromosome belonging to the genomic bases because interactions between chromosomes are very unlikely and (2) by downsampling evenly every 20 bases, and using different starting position each time - indeed it is well known in biology that there is a very strong correlation between close-by variants, approximately 10 kilo bases (kb) for *A. thaliana*, because

of evolutionary reasons. This enables us to get rid of redundancy between bases while looking for middle to long range interactions. We note that each genomic base is included in one and only one dataset, and each chromosome is split in 20 subdatasets.

**Combinatorial regulation of gene expression in breast cancer cells:** The breast cancer data set, as used in [10], includes $12,773$ genes classified into up-regulated or not up-regulated. Each gene is represented by 397 binary features which indicate the presence/absence of a sequence motif in the neighborhood of this gene. We aim to find combinations of motifs that are enriched in up-regulated genes.

### 2.3.2 Categories representative of population structure

***A. thaliana* GWAS** For the two datasets LY and *avrB*, we condition on the categorical covariate that is representative of the *population structure*; one can think of this covariate representing the subpopulation membership of a plant. We obtain this categorical covariate by running *k-means* on the first five principal components of the *kinship matrix* of the dataset, which represents the genetic relatedness of the plants [8, 9]. We select the number of clusters $k$ in a range from 2 to 8 that results in the best genomic inflation factor, a popular statistic in genetics for measuring the inflation of test statistics [6]. It results in $k = 3$ subpopulation clusters for *avrB* and $k = 5$ for LY. The *genomic inflation factor* is the ratio between the median of the distribution of the p-values of each tested feature combination and the median of the null distribution that is the $\chi^2$ distribution. Indeed, biologists manipulate large-p datasets, where most of the features are supposed to be independent to the label and only a few combinations bear the significant associated signal. If the ratio is $\gg 1$, then the genomic factor is inflated and the dataset is confounded, leading to many spurious associations. Conditioning on the correct covariates enables one to reduce the inflation factor by getting the $p$-value distribution closer to the null distribution.

**Combinatorial regulation of gene expression in breast cancer cells:** The covariate, with $k = 8$ and $k = 16$ categories, used with `FACS` was obtained by the same method as used with *A. thaliana*. Biologically, this means that we condition the analysis on groups of genes sharing similar motifs and try to find up-regulation associated motifs within these groups.

While one should acknowledge that it may be difficult to find biological justification for grouping the genes in this way in order to create categorical covariates, there is no natural alternative for this particular dataset, and the goal was to show that `FACS` can be handle real-world datasets with a relatively large number of categorical covariates.

### 2.4 How to evaluate correcting for confounders?

Validating the ability to correct for confounded patterns in real-world datasets is challenging, as the ground-truth is hardly ever known. In addition to the genomic inflation factor described above, a tool extensively used for that purpose when analyzing biological datasets are *Q-Q plots*. When applied to our setup, the x-axis corresponds to quantiles of the expected distribution of the $p$-values $p_S$, under the assumption that no pattern $\mathcal{S}$ is significantly associated. The $y$-axis corresponds to quantiles of the empirical distribution actually observed in our data sample. Since in practice most patterns will not be significantly associated, one expects the empirical distribution to approximately match the expected distribution. When this occurs, the plot approximately follow the identity $y = x$, with possible discrepancies towards the top-right part of the plot, corresponding to the few truly associated patterns. However, when confounding is present, the plots often look *inflated*, considerably deviating from the identity line upwards from the beginning of the plot.

### 2.4.1 Computing the inflation factor $\lambda$

The inflation factor $\lambda$ that is computed in the experimental analyses was first suggested by [6, p 1001] in the context of GWAS analysis (for single markers). A brief description of its computation is provided here, since it may not be well-known outside of the statistical genetics community.

First, statistics estimating the association of a marker with a phenotype are computed using the Cochran-Armitage [5, 2, 1] test. Under the null hypothesis of no association, these test statistics are approximately chi-squared distributed. Suppose there are $n$ markers, and the test statistics are

Figure 1: QQplot for the breast cancer dataset with the two methods, FACS and LAMP-$\chi^2$. The covariate is split into 16 categories.

$Y_1^2, Y_2^2, \ldots, Y_n^2$. Then, the inflation factor $\lambda$ is computed as

$$\lambda = \frac{\text{median}\{Y_1^2, Y_2^2, \ldots, Y_n^2\}}{0.675^2}.$$

The authors [6, p 1001] suggest that the term in the denominator enables a "robust" measure of $\lambda$, but no further justification was provided. However, we note that $0.675^2$ is close to the median of the chi-squared distribution (with one degree of freedom) $0.454936...$, which could be the origin of this value; in fact we use the more precise estimate of the median in our computation of $\lambda$.

In our analysis, the statistics $Y_i^2$ are computed for the all testable combinations. Although the Cochran-Mantel-Haenszel (CMH) test statistic is used here, rather than the Cochran-Armitage test, the CMH test statistic is also approximately chi-squared distributed under the null hypothesis, and so the $\lambda$ is computed in the same way.

It should be noted that [6] also suggests using the value of $\lambda$ to adjust the chi-squared statistics before computing the $p$-values; however, this practitioners often do not make this adjustment, since it does not change the *order* of the $p$-values. In this paper we follow this practice of not using $\lambda$ to adjust the statistics, but rather as only an indication of population structure.

Next, we describe the experimental setup specific to each of the two applications.

***A. thaliana* GWAS** The number of hits given in the main text for each plant disease is the total sum over all the chromosomes when taking all the genomic bases into account, i.e. all the 20 datasets per chromosome. The reported genomic inflation is the average over all the 20 datasets of all the chromosomes.

**Combinatorial regulation of gene expression in breast cancer cells:** Figure 1 shows Q-Q plots of the $p$-values for this dataset, demonstrating that FACS exhibits a improvement over LAMP-$\chi^2$. These results, in addition to the reduction in number of hits, indicate that it is necessary to tackle the inflation of test statistics in this setting and that conditioning on gene groups contributes to this goal. It is also apparent from the Q-Q plots that more p-value inflation remains in the Breast Cancer datasets than in the *Arabidopsis* dataset, which motivates future work in further reducing this inflation by correcting for additional covariates.

### 2.5   Summary of experimental results

We first discuss our experimental results on the biological datasets. In the case of *A. thaliana*, two representative datasets which exhibited a high amount of confounding were chosen [3], while the breast cancer dataset has been previously used in a significant pattern mining context [10].

Our results show that `FACS` can be applied to such real-world datasets, and can handle a moderate number of covariates (at least up to 16), while the simulation results, shown in Figures 1(a) and 1(b) in the main paper, confirms the result suggested by Theorem 2 that `FACS` has a runtime of the order $O(k \log k)$, and could potentially handle an even larger number of categories. Therefore, `FACS` should be able to process datasets with a higher number of categorical covariates with only a minor increase in computation time.

When analysing the real-world datasets, a preprocessing step that was introduced in the genetics literature [8, 9], and is based on principal component analysis, was used to obtain the categorical covariates. While it would be ideal to have datasets with pre-determined categorical covariates which are known to be highly confounded with the class labels, such datasets — with strong domain knowledge of the confounding mechanisms — were difficult to obtain. However, the simulation results shown in Figures 1(c) and 1(d) of the main paper indicate that, in situations where we are certain that there is confounding due to categorical covariates, `FACS` is able to account for such confounding and ignore combinations that are significantly associated with a covariate, while still finding the combinations that are truly associated with a phenotype.

As a final point, while the inflation factor $\lambda = 1.17$ and $\lambda = 1.21$ may seem high, this is not so uncommon for plant data where the populations are highly homogeneous. Furthermore, inflation factors of such a magnitude have been reported in meta-analyses of human height [11, Supp. Table 4]. While it may seem that one solution for lowering the inflation factor would be to consider additional categorical covariates, it is worth reiterating that a range of categories was considered and the number of categories corresponding to the lowest inflation factor was selected for these experiments. It is possible that the inflation factor is not closer to 1 rather due to the inherent population structure in these plant datasets and the low number of samples.

Overall, the results suggest that `FACS` is an important contribution in the development of significant pattern mining algorithms and hope that it will find application in the GWAS community. The code for `FACS` can be obtained from `https://github.com/BorgwardtLab/FACS`.