[Reviews · NeurIPS 2016]

Reviewer 1

Summary

The paper presents a significant discriminative items mining algorithm tailored for the identification of combination of features in the presence of categorical variables in classification problems. The approach is presented as an original improvement with respect to the state-of-the-art, as it takes into account confounding factors.

Qualitative Assessment

The paper presents in a clear and straightforward manner the Fast Automatic Conditional Search (FACS), a significant discriminative items mining algorithm that accounts for categorical covariates. The algorithm is designed to work on binary classification problems where the data is represented by binary features and where the number of samples is outnumbered by the number of features. While I have no significant comments on the first part of the work describing the method and algorithm, I have few remarks on the experiments and results. The synthetic data should be described more clearly. How many samples were considered? In which range did the small n large p scenario hold ? Runtime plots are too small, especially Figure 1 (c) and (d). Runtime evaluation should at least include a description of the computing resources adopted. For what concernes the real datasets, it is not clear how many samples there are in the A. thaliana dataset. In the Breast cancer gene expression dataset the small n large p scenario definitely does not hold. It may be worth presenting a different example. Moreover, this method would really be applicable if the binary feature condition could be relaxed and continuous feature could be taken into account.

Confidence in this Review

2-Confident (read it all; understood it all reasonably well)


Reviewer 2

Summary

This paper deals with the problem of identifying significant interactions between different feature combinations and a target variable while conditioning on categorical covariates (e.g. gender of the individuals in a study). This is particularly useful in cases where the test might be confounded by unmodeled factors. The main idea of the paper is to extend Tarone's concept of testability to account for the effect of categorical covariates. More in particular, this works present an extension of the Cochran-Mantel-Haenszel test that allows for the inclusion of categorical covariates.

Qualitative Assessment

This is an interesting paper that addresses shortcomings of existing approaches: 1) adjusts for multiple hypotheses testing without resorting to a Bonferroni correction 2) allows for the inclusion of categorical covariates I see two main issues with the method/paper: 1) the authors need to be more clear that the method proposed can only account for categorical covariates. This is sometimes stated in the paper, but lacks the space and clarity reserved to other statements in the paper. This is definitely a limitation that affects the usefulness of the method (and really shows in the experiments, where extremely unusual measures are taken to account for population structure, by running k-means on the first 5 principal components) 2) The experiments on real data are at best inconclusive. For example it's not clear that the number of hits reported in table 1 are the results on an increased sensitivity or specificity. The genomic control argument shows that the method as a somewhat better ability to control for type I errors, but a GC of 1.21 it's still quite inflated (especially with the LD pruning/averaging performed). Minor: - fonts in figure 1 are too small. - text in lines 279-185 doesn't seem to match the figure 1 legend

Confidence in this Review

2-Confident (read it all; understood it all reasonably well)


Reviewer 3

Summary

The paper introduces FACS, an items mining method that reduces the problem of confounding categorical variables. To achieve this, they combine the CMH test with Tarone’s testability criterion. They use Tarone’s criterion to get a set of testable feature subsets and CMH to test conditional independence between the feature combination and the class label given the covariates.

Qualitative Assessment

Overall, this is a good paper, introducing a novel algorithm, for a previously studied problem, with potentially high impact in genomics (and potentially other applications where confounding variables are an issue). The motivation of the paper could be improved by the authors presenting an example early in the introduction and stating what the feature combinations are, what the outcome is and what the confounding variables are. If there are any application domain papers that point out this, please indicate references or give some intuition as to how confounders become a problem. The experiments would be stronger if performed on a dataset for which some domain knowledge concerning significant features already exists. The knowledge would not need to be exhaustive (i.e. ground truth for all sets of features). Even information that some of the sets are definitely significant and others are definitely not significant could help evaluate the differences in precision/recall by FACS, LAMP and BONF-CMH.

Confidence in this Review

1-Less confident (might not have understood significant parts)


Reviewer 4

Summary

Given there are no significant discriminant itemset mining methods that assess associations between arbitrary combinations of features with a target variable while conditioning on an important covariate in order to account for the confounding (which substantively distorts the relationship between such feature combinations and the target, e.g., leading to false positives), the authors propose the novel FACS (Fast Automatic Conditional Search) algorithm to fill this methodological gap. FACS implements conditioned discriminant itemset mining with binary features, a binary target, and a k-level categorical conditioning covariate in O(k log k) time, while retaining relatively high statistical power. Specifically, FACS aims to find all feature subsets for which a conditional association test, namely the Cochran-Mantel-Haenszel test, rejects the null hypothesis that the set of feature combinations induced by some feature subset S is independent of the target Y, conditioned on the confounding covariate C, after correction for multiple testing based on Tarone’s testability criterion. Experimental results comparing FACS with up to 4 significant discriminant itemset mining methods using a simulated dataset and two real datasets suggest that FACS is relatively efficient, especially in cases where the conditioning covariate has many levels, and reduces false positive detection in the presence of confounding.

Qualitative Assessment

Confounding is a significant concern in detecting “true” statistical associations between features and a target variable and should be addressed appropriately in high-dimensional data contexts. Unfortunately, the confounding problem is frequently ignored in the development of new machine learning methods, which limits their real-world use in biomedical/biological settings. The authors propose an interesting solution to a gap in the field. To strengthen this result, the authors may want to consider revisiting the experimental analysis with real data. The performance evaluation metric appears to be counts of feature combination hits, which is not as informative as other metrics, e.g., accuracy.

Confidence in this Review

2-Confident (read it all; understood it all reasonably well)


Reviewer 5

Summary

This paper proposes a method to test all combinations of (binary) covariates for association with a (binary) label, which allows to correct for potential (categorical) confounding factors, and which is suited for high-dimensional settings. The method relies on the Cochran-Mantel-Haenszel-test (CMH) test. The challenge comes from the fact that the number of possible combinations of covariates is extremely high in a high-dimensional setting, resulting in a large multiple hypothesis testing problem. The authors tackle this problem by applying Tarone's testability criterion to the CMH test. Their contribution is the design of an algorithm that can perform such task in a computationally efficient manner.

Qualitative Assessment

This paper is technically sound, reads well and the contribution is well explained. I think the paper is incremental in that it brings together 2 existing blocks (CMH test and Tarone testability criterion) as it has been done before for the chisquare test with Tarone testability criterion. However bringing together CMH and Tarone is not trivial and the contribution (algorithm + proofs) is significant. I also have a comment on the results obtained with simulated data. The precision is lower for the proposed method than for LAMP-chisquare when the signal is rather weak. However, in real GWAS datasets, the signal strength is actually weak and therefore applying FACS instead of LAMP-chisquare on real datasets would according to the simulations lead to decreased statistical power. It would be interesting to discuss this a little bit more, explain what this implies for real datasets, and if possible why FACS-CMH is weaker than LAMP-chisquare for rather weak signals. I have spotted a few typos: l31: is to present l115: his -> this l166: with the

Confidence in this Review

2-Confident (read it all; understood it all reasonably well)


Reviewer 6

Summary

This paper proposed a discriminative itemset mining algorithm named FACS that deals with categorical co-founding effect in statistical analysis with specific applications in biology. FACS corrects for a confounding covariate in multiple hypothesis testing. The main contribution of FACS is to allow applying Tarone's testability criterion to the CMH test. At the same time, FACS is very fast and scalable.

Qualitative Assessment

The paper is well written and clearly organized, especially where it motivates the problem and the goals of the paper. It is a novel point of view to aim to find all feature subsets for which a statistical association test rejects the null hypothesis, and thus allowing to correct for a confounding categorical covariate. The authors results in the scope of the paper demonstrates that FACS keeps the computational efficiency, statistical power and the ability to correct for multiple hypothesis testing of existing method. The introduction of the branch-and-bound algorithm for conditional statistical association tests is novel and well-explained. Along with the results, the authors provide good intuition behind the methods and the key aspects, such as the appropriate testability criterion for the CMH test and the pruning criterion for the CMH test. So I think this work has provides a lot of new insights to field of machine learning in very high-dimensional settings. The paper has some minor shortcomings: 1) The proposed technique FACS is limited to applications with binary features with binary labels and cofounding factors. This setting is restricting its wide applications, although the extension of continuous setting is not trivial. 2) The experimental part is weak. In simulations, I would not agree with the authors' conclusion that Bonf-CMH is worse than FACS. Bonf-CMH actually performs well in terms of speed and false positive detection with good precision (Figure 1C). More insights could be put into the distinction between the two methods. Further, in real-world applications, FACS is only tested on small size GWAS dataset. I would strongly recommend authors to apply FACS on large scale GWAS data, where we have many cases and controls and large number of genes and report the effectiveness. On such a small datasets, it is really hard to validate the proposed algorithm.

Confidence in this Review

1-Less confident (might not have understood significant parts)